# DIFFUSION ADVERSARIAL REPRESENTATION LEARNING FOR SELF-SUPERVISED VESSEL SEGMENTATION

**Boah Kim**[*]**, Yujin Oh**[*]**, Jong Chul Ye**
Korea Advanced Institute of Science and Technology (KAIST), Daejeon, Republic of Korea
{boahkim,yujin.oh,jong.ye}@kaist.ac.kr

## ABSTRACT

Vessel segmentation in medical images is one of the important tasks in the diagnosis of vascular diseases and therapy planning. Although learning-based segmentation approaches have been extensively studied, a large amount of ground-truth labels are required in supervised methods and confusing background structures make neural networks hard to segment vessels in an unsupervised manner. To address this, here we introduce a novel diffusion adversarial representation learning (DARL) model that leverages a denoising diffusion probabilistic model with adversarial learning, and apply it to vessel segmentation. In particular, for self-supervised vessel segmentation, DARL learns the background signal using a diffusion module, which lets a generation module effectively provide vessel representations. Also, by adversarial learning based on the proposed switchable spatially-adaptive denormalization, our model estimates synthetic fake vessel images as well as vessel segmentation masks, which further makes the model capture vessel-relevant semantic information. Once the proposed model is trained, the model generates segmentation masks in a single step and can be applied to general vascular structure segmentation of coronary angiography and retinal images. Experimental results on various datasets show that our method significantly outperforms existing unsupervised and self-supervised vessel segmentation methods.

## 1 INTRODUCTION

In the clinical diagnosis of vascular diseases, vessel segmentation is necessary to analyze the vessel structures and therapy planning. In particular, when diagnosing coronary artery disease, X-ray angiography is taken to enhance vessel visualization by injecting a contrast agent into the blood vessels (Cong et al., 2015). However, it is challenging to extract vessels accurately due to low contrast, motion artifacts, many tiny branches, structural interference in the backgrounds, etc (Xia et al., 2019; Chen et al., 2014).

To segment vascular structures, various segmentation methods have been explored. Traditional optimization models (Law & Chung, 2009; Taghizadeh Dehkordi et al., 2014) typically require complicated preprocessing steps and manual tuning. Furthermore, they are computationally expensive to process many images. On the other hand, learning-based methods (Nasr-Esfahani et al., 2016; Fan et al., 2018; Chen et al., 2019) generate segmentation maps in real-time once the models are trained. However, supervised methods require a huge amount of labeled data for training, which complicates their use in practical applications. Also, existing unsupervised methods designed on natural images are difficult to apply to medical vessel images due to low contrast subtle branches and confusing background structures. Although a recent self-supervised method (Ma et al., 2021) is presented to learn vessel representations, this requires two different adversarial networks to segment vessels, which leads to increasing training complexity.

Recently, diffusion models such as denoising diffusion probabilistic model (DDPM) (Ho et al., 2020) has become one of the main research topics in modeling data distribution and sampling diverse images. By learning the Markov transformation of the reverse diffusion process from Gaussian

---

[*] co-first authors

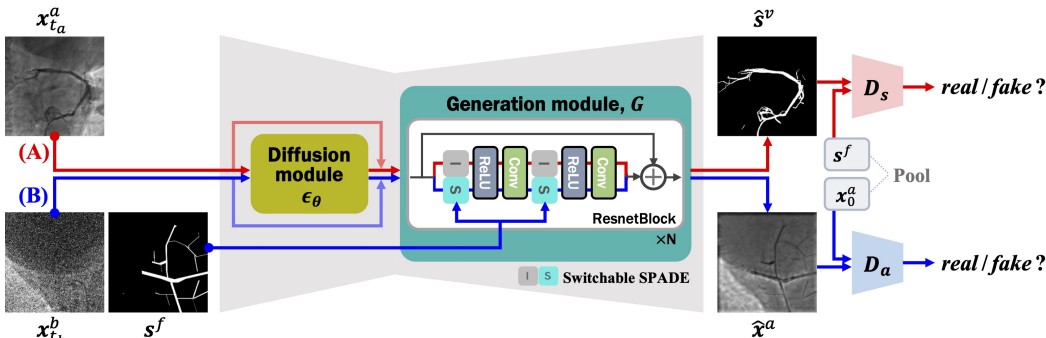

Figure 1: Our proposed diffusion adversarial representation model for self-supervised vessel segmentation. In path (A), given a real noisy angiography image $x_{t_a}^a$, our model estimates vessel segmentation masks $\hat{s}^v$. In path (B), given a real noisy background image $x_{t_b}^b$ and a vessel-like fractal mask $s^f$, our model generates a synthetic angiography image $\hat{x}^a$.

noise to data, DDPM is successfully applied to many low-level computer vision tasks such as super-resolution (Chung et al., 2022), inpainting (Lugmayr et al., 2022), and colorization (Song et al., 2020). For high-level vision tasks, while a recent study (Baranchuk et al., 2021) shows that DDPM can capture semantic information and be used as image representations, methods applying DDPM in learning semantic segmentation without labeled data have so far not been developed. Also, the sampling process of the diffusion models often takes a relatively long time.

In this paper, we introduce a novel concept of diffusion adversarial representation learning (DARL), which is a non-iterative version of the diffusion-based generative model and can be successfully applied to self-supervised vessel segmentation without ground-truth labels. As illustrated in Figure 1, our model is composed of a diffusion module and a generation module, which learns semantic information of vessels via adversarial learning. Specifically, based on the observation that the diffusion model estimates the noise added to the perturbed input data, and the adversarial learning model generates images for given the noisy vectors, we can naturally connect the diffusion model with the adversarial model. This allows our model not only to generate images in real time but also to segment vessels with robustness to noises and various modalities. Here, inspired by the spatially-adaptive denormalization (SPADE) layer (Park et al., 2019) that is effective in image synthesis given semantic masks, we present a *switchable* version of SPADE in the generation module to estimate vessel segmentation maps and mask-based fake angiograms simultaneously. This can yield a synergy effect in learning vessel representation by extracting proper features for angiogram synthesis.

More specifically, as shown in Figure 1, for given unpaired background images and angiography images that are taken before and after injection of the contrast agent, there are two paths for feeding the inputs into our proposed model: (A) when the real angiography images are given, our model without the SPADE estimates vessel segmentation maps; (B) when the background images are given, our model with the SPADE generates synthetic angiograms that composite vessel-like semantic masks with the input backgrounds. Also, as each vessel-like semantic mask in the (B) path can be regarded as the pseudo-label for the generated angiography image, by feeding the synthetic angiograms into the (A) path again, we apply the cycle consistency between the segmentation maps and the labels of fractal masks to capture semantic information of vessels. In addition, by designing the diffusion module to intensively learn the background signal, we let the module consider vessel structures of angiography images as outlier when estimating the latent feature. Thereby, vessel structures represented in the output of the diffusion module can guide the generation module to effectively segment the vessels.

We build our model on X-ray coronary angiography using XCAD dataset (Ma et al., 2021) and apply to several different blood vessel datasets, including retinal images. Experimental results show that our method outperforms several baseline methods by large margins for vessel segmentation tasks in the absence of labeled data. The main contributions are summarized as:

1. We propose a diffusion adversarial representation model, a non-iterative version of diffusion model for image generation, and apply it for self-supervised vessel segmentation.

Specifically, the latent features of our diffusion module provide vessel information and thus improve the segmentation performance.

2. Through the proposed generation module with switchable SPADE layers, our model not only generates synthetic angiography images but also segments vessel structures.

3. Experimental results verify that our model achieves superior segmentation performance by learning vessel representations. In particular, although the model is trained using X-ray coronary angiograms, it provides the state-of-the-art performance for un-/self-supervised retinal vessel segmentation as well, confirming the generalization capability of the model.

## 2 BACKGROUNDS AND RELATED WORKS

**Denoising diffusion probabilistic model** Diffusion model (Sohl-Dickstein et al., 2015; Ho et al., 2020; Song & Ermon, 2019) is one of generative models that sample realistic data by learning the distribution of real images. In particular, the denoising diffusion probabilistic model (DDPM) (Ho et al., 2020) with a score matching has been shown superior performance in image generation. Specifically, DDPM learns the Markov chain to convert the Gaussian noise distribution $x_T \sim \mathcal{N}(0, I)$ into the target distribution $x_0$. In the forward diffusion process, the noise is gradually added the noise to the data by:

$$q(x_t|x_{t-1}) = \mathcal{N}(x_t; \sqrt{1 - \beta_t}x_{t-1}, \beta_t I), \tag{1}$$

where $\beta_t \in [0, 1]$ is a fixed variance. Accordingly, a noisy target $x_t$ distribution from the data $x_0$ is represented as:

$$q(x_t|x_0) = \mathcal{N}(x_t; \sqrt{\alpha_t}x_0, (1 - \alpha_t)I), \tag{2}$$

where $\alpha_t = \Pi_{s=1}^t (1 - \beta_s)$. Then, DDPM is trained to approximate reverse diffusion process:

$$p_\theta(x_{t-1}|x_t) = \mathcal{N}(x_{t-1}; \mu_\theta(x_t, t), \sigma_t^2 I), \tag{3}$$

where $\sigma_t$ is a fixed variance, and $\mu_\theta$ is a parameterized mean with the noise predictor $\epsilon_\theta$:

$$\mu_\theta(x_t, t) = \frac{1}{\sqrt{1 - \beta_t}} \left( x_t - \frac{\beta_t}{\sqrt{1 - \alpha_t}}\epsilon_\theta(x_t, t) \right). \tag{4}$$

Thus, in the generative process, the sample can be obtained from the Gaussian noise by the iterative denoising steps: $x_{t-1} = \mu_\theta(x_t, t) + \sigma_t z$, where $z \sim \mathcal{N}(0, I)$.

Through this stochastic process, DDPM provides diverse realistic samples and has been exploited in many applications, including super-resolution (Chung et al., 2022; Saharia et al., 2021), inpainting (Lugmayr et al., 2022), and colorization (Song et al., 2020; Saharia et al., 2022). However, the application study of semantic segmentation is limited. Although several works (Baranchuk et al., 2021; Amit et al., 2021) are recently presented to solve high-level vision problems, they require annotated data to train the models.

**Self-supervised vessel segmentation** For the vessel segmentation task, it is difficult to obtain fine-grained labels for supervised learning, since the vessel has complex structures with numerous tiny branches. While this label scarcity issue can be alleviated by semi- or unsupervised learning, fully unsupervised methods to segment the tiny vessels with reasonable performance are relatively scarce. In fact, recent unsupervised learning methods trained with natural images have great generalization capability on unseen datasets (Ahn et al., 2021; Chen et al., 2019; Melas-Kyriazi et al., 2022), thus they can be easily adapted to medical image segmentation tasks. However, due to the unique characteristics of angiography, e.g. confusing background factors and sophisticated vessel structures, any unsupervised methods designed for natural image segmentation may get degraded performance when they are applied to vessel segmentation of noisy angiography images. As a type of unsupervised learning, self-supervised learning also has been introduced to utilize self-generated supervisory labels from data themselves to efficiently learn target representations in various medical image segmentation tasks and has demonstrated its potential (Mahmood et al., 2019; Ma et al., 2021; Oh & Ye, 2021). Specifically, Ma et al. (2021) introduces an end-to-end adversarial learning framework for vessel segmentation with the CycleGAN (Zhu et al., 2017) structure, which learns realistic angiogram generation that adds fractal-guided pseudo labels to the background images. However, the simple arithmetic operation for synthetic vessel generation often fails to yield realistic pseudo-vessel images, thus training the adversarial networks using unrealistic synthetic images is difficult to produce optimal segmentation performance.

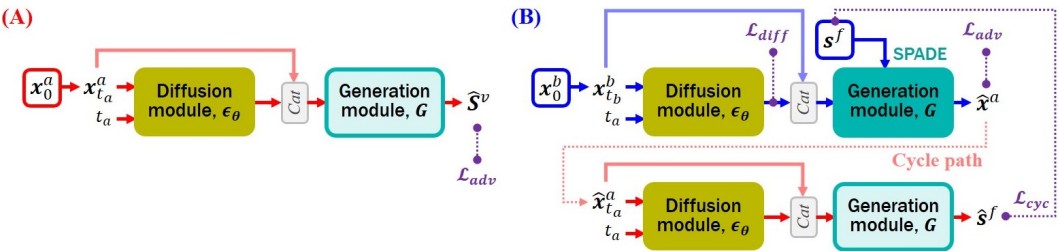

Figure 2: Training flow of our model. The generation module $G$ with the switchable SPADE layers takes $\epsilon_{\theta}$ and the noisy images, and generates desired outputs corresponding to the paths. $x_{t_a}^a$ and $x_{t_b}^b$ denote the noisy angiography and background images, where $t_a$ and $t_b$ are noise schedules. $\hat{s}^v$ is the generated vessel segmentation, and $\hat{x}^a$ is the synthetic angiography images. $s^f$ is the vessel-like fractal masks. $Cat$ denotes the concatenation of images in channel dimension. $\mathcal{L}_{diff}$, $\mathcal{L}_{adv}$, and $\mathcal{L}_{cyc}$ are the diffusion loss, the adversarial loss, and the cycle loss, respectively.

## 3 DIFFUSION ADVERSARIAL REPRESENTATION LEARNING

In this section, we describe our novel diffusion adversarial representation learning (DARL) model, tailored for self-supervised vessel segmentation. We call the images before and after injecting the contrast agents into the blood vessels as *background* and *angiography*, respectively. Note that due to the different scanning times, these two images have different contrasts and are not aligned, caused by the movements of patients. Thus, as shown in Figure 1, our DARL model is trained on unpaired angiography images $x_0^a$ and background images $x_0^b$.

Specifically, our model is comprised of a diffusion module $\epsilon_{\theta}$ to estimate latent features, a generation module $G$ to estimate both the vessel segmentation masks $\hat{s}^v$ and the synthetic angiograms $\hat{x}^a$, and two discriminators $(D_s, D_a)$ to distinguish real and fake images of the vessel masks and the angiograms, respectively. Here, in generating angiography images, we provide the vessel-like fractal masks $s^f$ presented by Ma et al. (2021) to the generation module to perform image synthesis based on semantic layouts. Moreover, to estimate the segmentation maps and angiography images effectively, we design the generation module with novel switchable SPADE layers, where the SPADE Park et al. (2019) facilitates the semantic image synthesis.

**Generation module with switchable SPADE layers** As illustrated in Figure 1, the proposed generation module consists of $N$ residual blocks (ResnetBlock) that have switchable SPADE (S-SPADE) layers. Note that the (A) and (B) paths of our model are implemented simultaneously by sharing the learnable parameters except for the S-SPADE layers. Then, let $v \in \mathbb{R}^{M \times C \times H \times W}$ be the feature map in the ResnetBlock, where $M$, $C$, $H$, and $W$ are the size of batch, channel, height, and width, respectively. The switchable SPADE layer normalizes feature maps differently depending on the existence of an input mask $s$:

$$v = \begin{cases} \text{SPADE}(v, s), & \text{if mask } s \text{ is given,} \\ \text{IN}(v), & \text{otherwise,} \end{cases} \tag{5}$$

where IN is the instance normalization (Ulyanov et al., 2017). So, when our model is given the fake vessel mask $s^f$, the SPADE is computed by:

$$x_{m,c,h,w} = \gamma_{c,h,w}(s^f) \frac{x_{m,c,h,w} - \mu_c}{\sigma_c} + \beta_{c,h,w}(s^f), \tag{6}$$

where $x_{m,c,h,w}$ denotes the $(m, c, h, w)$-th element of the feature tensor $v$, $(\mu_c, \sigma_c)$ are the mean and standard deviation of the feature map in channel $c$, and $(\gamma_{c,h,w}, \beta_{c,h,w})$ are learned modulation parameters during training.

Thus, in the (A) path, given the noisy angiogram $x_t^a$ and the latent feature $\epsilon_{\theta}(x_t^a, t)$, the generation module $G$ estimates the vessel segmentation masks $\hat{s}^v$ without the SPADE:

$$\hat{s}^v = G(\epsilon_{\theta}(x_t^a, t); \mathbf{0}). \tag{7}$$

On the other hand, in the (B) path that provides the fractal mask $s^f$, the generation module taking the noisy background $x_t^b$ and its latent feature $G(\epsilon_{\theta}(x_t^b, t))$ synthesizes the fake angiograms $\hat{x}^a$:

$$\hat{x}^a = G(\epsilon_{\theta}(x_t^b, t); s^f). \tag{8}$$

### 3.1 NETWORK TRAINING

In contrast to the DDPM that pretrains the diffusion model, our method trains the diffusion module, generation module, and discriminators simultaneously using an adversarial learning framework. Figure 2 depicts the detailed training flow of our model. There are two distinct paths: (A) one feeds the real angiograms $x_0^a$ into the model to provide vessel masks $\hat{s}^v$, and (B) the other takes the real backgrounds $x_0^b$ and the fractal masks $s^f$ for the model to generate fake angiograms $\hat{x}^a$. Here, as shown in Figure 2(B), since the input fractal masks can be regarded as vessel segmentation labels of the fake angiograms, we forward the fake angiograms generated through the (B) path to the (A) path, and apply cycle consistency between the estimated segmentation masks and the fractal masks to capture the vessel information.

#### 3.1.1 LOSS FUNCTION

To train the model, we employ LSGAN (Mao et al., 2017) framework, which leads to the alternating application of the following two optimization problems:

$$\min_{\boldsymbol{\theta},G}\mathcal{L}^G(\epsilon_{\boldsymbol{\theta}},G,D_s,D_a), \quad \min_{D_s,D_a}\mathcal{L}^D(\epsilon_{\boldsymbol{\theta}},G,D_s,D_a), \tag{9}$$

where $\mathcal{L}^G$, and $\mathcal{L}^D$ denotes the losses for the diffusion/generator and discriminator, respectively, which are given by:

$$\mathcal{L}^G(\epsilon_{\boldsymbol{\theta}},G,D_s,D_a) = \mathcal{L}_{diff}(\epsilon_{\boldsymbol{\theta}}) + \alpha\mathcal{L}_{adv}^G(\epsilon_{\boldsymbol{\theta}},G,D_s,D_a) + \beta\mathcal{L}_{cyc}(\epsilon_{\boldsymbol{\theta}},G), \tag{10}$$

$$\mathcal{L}^D(\epsilon_{\boldsymbol{\theta}},G,D_s,D_a) = \mathcal{L}_{adv}^{D_s}(\epsilon_{\boldsymbol{\theta}},G,D_s) + \mathcal{L}_{adv}^{D_a}(\epsilon_{\boldsymbol{\theta}},G,D_a), \tag{11}$$

where $\alpha$ and $\beta$ are hyperparameters, $\mathcal{L}_{diff}$ is the diffusion loss, $\mathcal{L}_{adv}$ is adversarial loss, and $\mathcal{L}_{cyc}$ is cyclic reconstruction loss. The detailed description of each loss function is as follows.

**Diffusion loss**  Recall that the diffusion module learns the distribution of images to estimate meaningful latent features of the inputs. We follow the standard loss for DDPM training (Ho et al., 2020):

$$\mathcal{L}_{diff}(\epsilon_{\boldsymbol{\theta}}) := \mathbb{E}_{t,\boldsymbol{x}_0,\epsilon}\Big[\|\epsilon - \epsilon_{\theta}(\sqrt{\alpha_t}\boldsymbol{x}_0 + \sqrt{1-\alpha_t}\epsilon, t)\|^2\Big]. \tag{12}$$

where $\epsilon \sim \mathcal{N}(\mathbf{0},\boldsymbol{I})$. In particular, to let the diffusion module represent the vessels of angiograms effectively, we define the diffusion loss on the background images, i.e. $\boldsymbol{x}_0 = \boldsymbol{x}_0^b$ in the (B) path and set the sampling schedule in $t \in [0,T]$. Accordingly, the diffusion module is trained intensively to learn the background image signal, allowing the module in the (A) path to regard the vessel structures of the angiograms as outlier and represent vessels in the latent features.

**Adversarial loss**  To generate both vessel segmentation masks and synthetic angiograms without the ground-truth labels, the proposed model is trained by adversarial learning using the two discriminators $D_s$ and $D_a$. As shown in Figure 1, the discriminator $D_s$ attempts to distinguish the estimated segmentation masks $\hat{s}^v$ from the real fractal mask $s^f$ (in the (A) path), while the discriminator $D_a$ tries to discriminate between the generated angiograms $\hat{x}^a$ and the real aniography images $x_0^a$ (in the (B) path). As we employ LSGAN (Mao et al., 2017), the adversarial loss of generator $\mathcal{L}_{adv}^G$ can be formulated by:

$$\mathcal{L}_{adv}^G(\epsilon_{\boldsymbol{\theta}},G,D_s,D_a) = \mathbb{E}_{\boldsymbol{x}^a}[(D_s(G(\epsilon_{\boldsymbol{\theta}}(\boldsymbol{x}^a);\mathbf{0}))-1)^2] + \mathbb{E}_{\boldsymbol{x}^a,\boldsymbol{s}^f}[(D_a(G(\epsilon_{\boldsymbol{\theta}}(\boldsymbol{x}^b);\boldsymbol{s}^f))-1)^2]. \tag{13}$$

On the other hand, the discriminators are trained to compete against the generator with the adversarial loss functions, $\mathcal{L}_{adv}^{D_s}$ and $\mathcal{L}_{adv}^{D_a}$, which are defined by:

$$\mathcal{L}_{adv}^{D_s}(\epsilon_{\boldsymbol{\theta}},G,D_s) = \frac{1}{2}\mathbb{E}_{\boldsymbol{s}^f}[(D_s(\boldsymbol{s}^f)-1)^2] + \frac{1}{2}\mathbb{E}_{\boldsymbol{x}^a}[(D_s(G(\epsilon_{\boldsymbol{\theta}}(\boldsymbol{x}^a);\mathbf{0})))^2], \tag{14}$$

$$\mathcal{L}_{adv}^{D_a}(\epsilon_{\boldsymbol{\theta}},G,D_a) = \frac{1}{2}\mathbb{E}_{\boldsymbol{x}_0^a}[(D_a(\boldsymbol{x}_0^a)-1)^2] + \frac{1}{2}\mathbb{E}_{\boldsymbol{x}^a,\boldsymbol{s}^f}[(D_a(G(\epsilon_{\boldsymbol{\theta}}(\boldsymbol{x}^b);\boldsymbol{s}^f)))^2]. \tag{15}$$

This adversarial loss enables the single generator $G$ to fool the discriminator $D_s$ and $D_a$, by generating realistic segmentation masks $\hat{s}^v = G(\epsilon_{\boldsymbol{\theta}}(\boldsymbol{x}^a);\mathbf{0})$ and angiograms $\hat{x}^a = G(\epsilon_{\boldsymbol{\theta}}(\boldsymbol{x}^b);\boldsymbol{s}^f)$. In contrast, the discriminators attempt to distinguish these generated images being fake and the real images of $\boldsymbol{s}^f$ and $\boldsymbol{x}_0^a$ being real.

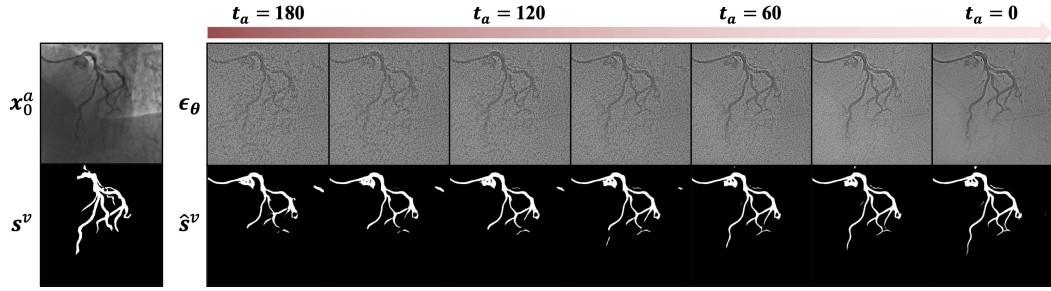

Figure 3: Vessel segmentation according to the noise level $t_a$. Our model estimates the segmentation masks $\hat{s}^v$ using the latent features $\epsilon_\theta$ for the noisy angiograms $x_{t_a}^a$. $s^v$ is the ground-truth label.

**Cyclic reconstruction loss** For the generator $G$ to capture the semantic information of the vessels, we also constrain our model with the cyclic reconstruction loss on the fractal masks. Specifically, as the vessel-like fractal masks $s^f$ can be labels for the synthetic angiograms $\hat{x}^a$ generated in the (B) path, we feed the $\hat{x}^a$ into our model and reconstruct the fractal masks by the (A) path. Therefore, the cyclic reconstruction loss is computed between the reconstructed segmentation masks and the real fractal masks, which can be written by:

$$\mathcal{L}_{cyc}(\epsilon_\theta, G) = \mathbb{E}_{x^b, s^f}[||G(\epsilon_\theta(G(\epsilon_\theta(x^b); s^f)); \mathbf{0}) - s^f||_1]. \tag{16}$$

Here, we solve the segmentation problem as a vessel mask image generation, which is why we use L1 loss in the cyclic loss.

### 3.1.2 IMAGE PERTURBATION FOR THE MODEL INPUT

Given real images of $x_0^a$ and $x_0^b$, our diffusion module takes noisy angiograms $x_{t_a}^a$ in the (A) path and noisy background images $x_{t_b}^b$ in the (B) path as the input, in which each noisy image is sampled based on the forward diffusion process (2):

$$x_t = \sqrt{\alpha_t} x_0 + \sqrt{1 - \alpha_t} \epsilon, \tag{17}$$

where $\epsilon \sim \mathcal{N}(\mathbf{0}, I)$, and both of $t_a$ and $t_b$ are uniformly sampled time step in $[0, T]$. Here, for the diffusion module not only to learn the background image signal in the (B) path but also to provide useful information for the generation module to segment the vessel structures under even certain noisy angiogram images in the (A) path, we sample $t_a$ in the range of $[0, T_a]$ where $T_a < T$. Empirically, we found that this makes our model learn vessel representations robust to the noise.

### 3.2 INFERENCE OF VESSEL SEGMENTATION

The inference phase of DARL is different from the conventional diffusion model in that our model do not require iterative reverse process, similar to the recent diffusion-based unsupervised learning method called DiffuseMorph (Kim et al., 2021). Specifically, once the proposed DARL is trained, in the inference, we can obtain the vessel segmentation masks of angiograms from the (A) path by one step. For the noisy angiograms $x_{t_a}^a$ given by the forward diffusion process (17), our model provides the vessel segmentation masks using the latent features $\epsilon_\theta(x_{t_a}^a, t_a)$ estimated from the diffusion module. As shown in Figure 3, our model can generate the segmentation masks for any noise level $t_a$ within a certain range (i.e. $[0, T_a]$). Nevertheless, since the angiography image $x_0^a$ can be considered as one of the clean target images, the closer $t_a$ is to zero, the better the vessel segmentation performance. Therefore, we test our model by setting $t_a = 0$.

## 4 EXPERIMENTS

In this section, we thoroughly evaluate the vessel segmentation performance of our method. We firstly compare the proposed DARL to existing unsupervised and self-supervised baseline models on various angiography datasets, including X-ray coronary angiography and retinal images. Also, we study the noise robustness of our model. Then, we analyze the success of our model in vessel representation and conduct an ablation study.

**Datasets** To realize the self-supervised learning framework, we train our model with the publicly available unlabeled X-ray coronary angiography disease (XCAD) dataset obtained during stent placement surgery and generated synthetic fractal masks (Ma et al., 2021). When training the network, each angiography and background data is independently sampled. Also, in testing, we utilize external 134 XCA (Cervantes-Sanchez et al., 2019) and 30 XCA (Hao et al., 2020) datasets. Furthermore, we evaluate cross-organ generalization capability on retinal imaging datasets; DRIVE (Staal et al., 2004) and STARE (Hoover & Goldbaum, 2003). Details of the datasets are in Appendix B.

**Implementation details** Our model is implemented by employing the network architectures proposed in DDPM (Ho et al., 2020) and SPADE (Park et al., 2019) for the diffusion module and the generation module, respectively. Also, for the discriminators, we use the network of PatchGAN (Isola et al., 2017). To train the model, we set the number of time steps as $T = 2000$ with the linearly scheduled noise levels from $10^{-6}$ to $10^{-2}$. Within this range, we sample the noisy angiograms by setting $T_a$ to 200. Also, we set the hyperparameters of loss function as $\alpha = 0.2$ and $\beta = 5$. Our model is optimized by using the Adam algorithm (Kingma & Ba, 2014) with a learning rate of $5 \times 10^{-6}$ on a single GPU card of Nvidia Quadro RTX 6000. We train the model for 150 epochs, and the model in the epoch with the best performance on the validation set is used for test data. All the implementations are done using the library of PyTorch (Paszke et al., 2019) in Python. The details of network structures and hyperparameter setting can be found in Appendix.

**Baseline methods and metrics** We compare our model to several baseline methods of un-/self-supervised learning, which do not require ground-truth vessel labels. For unsupervised learning methods, we utilize Spatial-Guided Clustering (SGC) (Ahn et al., 2021), Redrawing (Chen et al., 2019), and Deep Spectral (DS) (Melas-Kyriazi et al., 2022). For self-supervised learning methods, we employ Self-supervised Transformer with Energy-based Graph Optimization (STEGO) (Hamilton et al., 2022), Deep Adversarial (DA) (Mahmood et al., 2019), and Self-Supervised Vessel Segmentation (SSVS) (Ma et al., 2021). All these methods are implemented under identical training conditions to our model, unless the method needs no training procedure. For baseline methods that

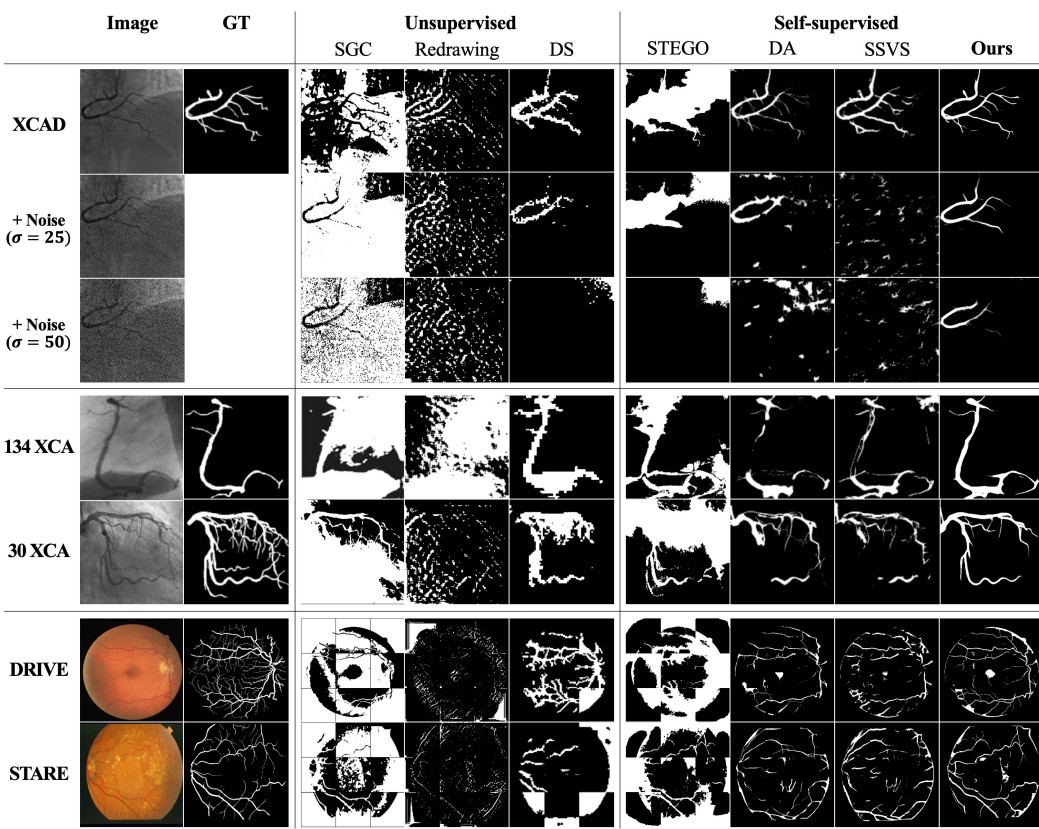

Figure 4: Visual comparison results on the vessel segmentation of various angiography images.

Table 1: Quantitative evaluation results on the vessel segmentation of various angiography images.

| Data | Metric | Unsupervised | | | Self-supervised | | | |
|---|---|---|---|---|---|---|---|---|
| | | SGC | Redrawing | DS | STEGO | DA | SSVS | **Ours** |
| XCAD | IoU | $0.060_{\pm0.034}$ | $0.059_{\pm0.032}$ | $0.366_{\pm0.105}$ | $0.146_{\pm0.070}$ | $0.375_{\pm0.066}$ | $0.410_{\pm0.087}$ | $\mathbf{0.471_{\pm0.076}}$ |
| | Dice | $0.111_{\pm0.060}$ | $0.109_{\pm0.056}$ | $0.526_{\pm0.131}$ | $0.249_{\pm0.103}$ | $0.542_{\pm0.073}$ | $0.575_{\pm0.091}$ | $\mathbf{0.636_{\pm0.072}}$ |
| | Precision | $0.062_{\pm0.034}$ | $0.139_{\pm0.081}$ | $0.469_{\pm0.127}$ | $0.152_{\pm0.077}$ | $0.557_{\pm0.115}$ | $0.590_{\pm0.119}$ | $\mathbf{0.701_{\pm0.115}}$ |
| **External test: Coronary angiography** | | | | | | | | |
| 134 XCA | IoU | $0.045_{\pm0.035}$ | $0.056_{\pm0.018}$ | $0.256_{\pm0.110}$ | $0.134_{\pm0.081}$ | $0.190_{\pm0.155}$ | $0.318_{\pm0.128}$ | $\mathbf{0.426_{\pm0.059}}$ |
| | Dice | $0.085_{\pm0.063}$ | $0.105_{\pm0.033}$ | $0.394_{\pm0.159}$ | $0.228_{\pm0.109}$ | $0.291_{\pm0.217}$ | $0.468_{\pm0.156}$ | $\mathbf{0.595_{\pm0.058}}$ |
| | Precision | $0.047_{\pm0.036}$ | $0.058_{\pm0.019}$ | $0.280_{\pm0.123}$ | $0.136_{\pm0.088}$ | $0.506_{\pm0.201}$ | $0.592_{\pm0.125}$ | $\mathbf{0.781_{\pm0.118}}$ |
| 30 XCA | IoU | $0.083_{\pm0.039}$ | $0.048_{\pm0.022}$ | $0.339_{\pm0.086}$ | $0.191_{\pm0.072}$ | $0.298_{\pm0.109}$ | $0.324_{\pm0.146}$ | $\mathbf{0.427_{\pm0.184}}$ |
| | Dice | $0.150_{\pm0.064}$ | $0.091_{\pm0.040}$ | $0.499_{\pm0.113}$ | $0.314_{\pm0.100}$ | $0.447_{\pm0.148}$ | $0.468_{\pm0.193}$ | $\mathbf{0.572_{\pm0.205}}$ |
| | Precision | $0.090_{\pm0.041}$ | $0.144_{\pm0.074}$ | $0.525_{\pm0.130}$ | $0.200_{\pm0.081}$ | $0.612_{\pm0.174}$ | $0.613_{\pm0.212}$ | $\mathbf{0.729_{\pm0.152}}$ |
| **Cross-modality test: Retinal imaging** | | | | | | | | |
| DRIVE | IoU | $0.063_{\pm0.055}$ | $0.057_{\pm0.033}$ | $0.217_{\pm0.143}$ | $0.152_{\pm0.073}$ | $0.245_{\pm0.090}$ | $0.314_{\pm0.101}$ | $\mathbf{0.372_{\pm0.148}}$ |
| | Dice | $0.115_{\pm0.093}$ | $0.105_{\pm0.059}$ | $0.333_{\pm0.201}$ | $0.257_{\pm0.106}$ | $0.386_{\pm0.117}$ | $0.469_{\pm0.119}$ | $\mathbf{0.525_{\pm0.161}}$ |
| | Precision | $0.069_{\pm0.061}$ | $0.199_{\pm0.155}$ | $0.243_{\pm0.175}$ | $0.169_{\pm0.100}$ | $0.503_{\pm0.218}$ | $0.549_{\pm0.216}$ | $\mathbf{0.617_{\pm0.271}}$ |
| STARE | IoU | $0.055_{\pm0.045}$ | $0.074_{\pm0.048}$ | $0.180_{\pm0.141}$ | $0.125_{\pm0.076}$ | $0.237_{\pm0.122}$ | $0.311_{\pm0.148}$ | $\mathbf{0.368_{\pm0.191}}$ |
| | Dice | $0.101_{\pm0.077}$ | $0.134_{\pm0.080}$ | $0.281_{\pm0.201}$ | $0.216_{\pm0.109}$ | $0.367_{\pm0.167}$ | $0.454_{\pm0.185}$ | $\mathbf{0.508_{\pm0.216}}$ |
| | Precision | $0.058_{\pm0.047}$ | $0.227_{\pm0.157}$ | $0.205_{\pm0.172}$ | $0.135_{\pm0.092}$ | $0.427_{\pm0.233}$ | $0.490_{\pm0.230}$ | $\mathbf{0.537_{\pm0.280}}$ |

require heuristic thresholds, optimal performance is achieved by selecting data-specific thresholds within the range from 0.2 to 0.8 in increments of 0.1. To quantitatively evaluate the segmentation performance, we compute Intersection over Union (IoU), Dice similarity coefficient, and Precision.

## 4.1 EXPERIMENTAL RESULTS

Figure 4 shows the vessel segmentation masks from the baseline methods and our proposed method on three different coronary angiography datasets and two retinal imaging datasets. Quantitative evaluation results of the methods are presented in Table 1. The analysis of the results is as follows.

**Comparison of ours to baselines** When we compare the proposed method to the baselines, our model segments vessel structures including tiny branches more accurately. Also, as shown in Table 1, our model consistently achieves the SOTA performance by large margin compared to existing unsupervised and self-supervised methods. In specific, our network shows significantly improved precision scores, which demonstrates advantages of our DARL that effectively differentiates foreground vessel structure and eliminates false positive signals from the noisy backgrounds.

**Generalization capability** To verify that our trained DARL can be generally used for various vessel images taken from different machines or different anatomic region-of-interests (ROI), we evaluate the generalization capability by applying the models only trained on the XCAD dataset to the other datasets. First, for the external 134 XCA and 30 XCA datasets which have different resolutions and noise distributions to those of the XCAD dataset, as shown in Figure 4 and Table 1, our model achieves higher performance than the others. Also, with the DRIVE and STARE retinal datasets that have unseen data distributions due to the different modalities from the XCAD, our DARL shows the most promising cross-organ generalization performance. This may come from the proposed framework that reuses the generated angiography images for the segmentation process through the cycle path, diversifying the input data distribution. Also, the diffusion module learning the stochastic diffusion process enables our model to be used in general for vessel segmentation.

**Robustness to noises** As X-ray images are often acquired under low-dose radiation exposure to reduce potential risks, we further evaluate the performance of our model on simulated noisy angiograms. Using the XCAD dataset, we add Gaussian noise to the angiogram with different levels of $\sigma =10$, 25, and 50. We show the segmentation results according to the noise levels in Figure 4. Also, we report the quantitative evaluation results in Table 2. It is noteworthy that our DARL is the only method to segment vessel structures with reasonable performance under noise corruption. Since the proposed segmentation method is trained through the diffusion module that perturbs the input images, the model is highly robust to segment vessel structure even from the noisy data.

**Latent representation** To study the origin of the performance improvement, in Figure 5, we show the latent features $\epsilon_{\boldsymbol{\theta}}(\boldsymbol{x}_t, t)$ given $\boldsymbol{x}_0$ for (A) the angiography $\boldsymbol{x}_0 = \boldsymbol{x}_0^a$ and (B) the

Table 2: Results of noise robustness test according to the Gaussian noise with $\sigma$.

| $\sigma$ | Metric | Unsupervised | | | Self-supervised | | | |
|---|---|---|---|---|---|---|---|---|
| | | SGC | Redrawing | DS | STEGO | DA | SSVS | **Ours** |
| 10 | IoU | $0.066_{\pm0.033}$ | $0.052_{\pm0.031}$ | $0.331_{\pm0.104}$ | $0.144_{\pm0.073}$ | $0.353_{\pm0.065}$ | $0.258_{\pm0.079}$ | $\mathbf{0.451_{\pm0.080}}$ |
| | Dice | $0.122_{\pm0.059}$ | $0.096_{\pm0.053}$ | $0.487_{\pm0.133}$ | $0.245_{\pm0.107}$ | $0.519_{\pm0.073}$ | $0.404_{\pm0.099}$ | $\mathbf{0.617_{\pm0.076}}$ |
| | Precision | $0.069_{\pm0.035}$ | $0.126_{\pm0.077}$ | $0.480_{\pm0.135}$ | $0.157_{\pm0.091}$ | $0.481_{\pm0.104}$ | $0.477_{\pm0.117}$ | $\mathbf{0.710_{\pm0.115}}$ |
| 25 | IoU | $0.069_{\pm0.035}$ | $0.036_{\pm0.021}$ | $0.232_{\pm0.094}$ | $0.118_{\pm0.064}$ | $0.247_{\pm0.072}$ | $0.059_{\pm0.033}$ | $\mathbf{0.389_{\pm0.088}}$ |
| | Dice | $0.128_{\pm0.061}$ | $0.069_{\pm0.039}$ | $0.366_{\pm0.132}$ | $0.206_{\pm0.095}$ | $0.391_{\pm0.092}$ | $0.109_{\pm0.058}$ | $\mathbf{0.554_{\pm0.092}}$ |
| | Precision | $0.072_{\pm0.036}$ | $0.095_{\pm0.058}$ | $0.446_{\pm0.159}$ | $0.144_{\pm0.115}$ | $0.371_{\pm0.106}$ | $0.149_{\pm0.082}$ | $\mathbf{0.727_{\pm0.119}}$ |
| 50 | IoU | $0.070_{\pm0.025}$ | $0.020_{\pm0.012}$ | $0.077_{\pm0.065}$ | $0.060_{\pm0.050}$ | $0.102_{\pm0.056}$ | $0.021_{\pm0.013}$ | $\mathbf{0.269_{\pm0.081}}$ |
| | Dice | $0.130_{\pm0.045}$ | $0.040_{\pm0.022}$ | $0.136_{\pm0.109}$ | $0.108_{\pm0.088}$ | $0.180_{\pm0.091}$ | $0.041_{\pm0.025}$ | $\mathbf{0.417_{\pm0.101}}$ |
| | Precision | $0.072_{\pm0.026}$ | $0.061_{\pm0.038}$ | $0.221_{\pm0.168}$ | $0.076_{\pm0.067}$ | $0.169_{\pm0.094}$ | $0.060_{\pm0.038}$ | $\mathbf{0.716_{\pm0.147}}$ |

backgrounds $x_0 = x_0^b$ with $t = 100$, respectively. In contrast to the (B) path, the latent representation in the (A) path emphasizes the vessel structures. This implies that although there are no ground-truth labels, our model learns the background image representation so that the vessel structure can be captured as outlier, leading to improved segmentation performance.

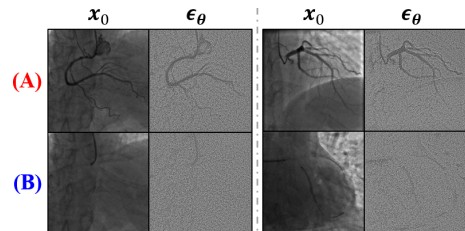

Figure 5: Estimated latent features $\epsilon_\theta$ in the (A) and (B) paths of our model.

**Ablation study**   Table 3 shows the evaluation results of several ablation studies. Implementation details and visual results are in Appendix D.1. (a) Our model without the diffusion module and $\mathcal{L}_{diff}$ shows lower performance by about 2% for all metrics compared to our model, which suggests that the diffusion module guides the generation module to extract vessel representation accurately. (b) The generation module without the proposed S-SPADE layers is degraded by more than 1% over (a) for all metrics, verifying that our SPADE-based unified generator effectively captures vessel semantic information through the synergy of learning both image segmentation and generation. (c) Through the implementation of our model without the proposed cyclic loss $\mathcal{L}_{cyc}$, we verify that $\mathcal{L}_{cyc}$ allows our model to segment proper vessel regions. (d) When training our model by converting the L1 loss for $\mathcal{L}_{cyc}$ to the cross-entropy (CE) loss, the performance is much worse than ours in all metrics, which implies that our approach using L1 loss for the cycle path is proper to obtain the vessel masks.

Table 3: Results of ablation study on the proposed model and loss function.

| Method | Module | | Loss function | | | Metric | | |
|---|---|---|---|---|---|---|---|---|
| | Diffusion | Generation | $\mathcal{L}_{diff}$ | $\mathcal{L}_{adv}$ | $\mathcal{L}_{cyc}$ | IoU | Dice | Precision |
| Ours | ✓ | ✓ | ✓ | ✓ | ✓ | $\mathbf{0.471_{\pm0.076}}$ | $\mathbf{0.636_{\pm0.072}}$ | $\mathbf{0.701_{\pm0.115}}$ |
| (a) | | ✓ | | ✓ | ✓ | $0.449_{\pm0.077}$ | $0.616_{\pm0.074}$ | $0.646_{\pm0.106}$ |
| (b) | | w/o S-SPADE | | ✓ | ✓ | $0.439_{\pm0.080}$ | $0.606_{\pm0.080}$ | $0.620_{\pm0.111}$ |
| (c) | ✓ | ✓ | ✓ | ✓ | | $0.322_{\pm0.055}$ | $0.485_{\pm0.064}$ | $0.580_{\pm0.112}$ |
| (d) | ✓ | ✓ | ✓ | ✓ | L1→CE | $0.346_{\pm0.084}$ | $0.508_{\pm0.094}$ | $0.672_{\pm0.147}$ |

## 5   CONCLUSION

We present a non-iterative diffusion model called DARL for self-supervised vessel segmentation. Our model composed of the diffusion and generation modules learns vessel representation without labels via adversarial learning, in the guidance of latent features estimated from the diffusion module. Also, through the proposed switchable SPADE layer, we generate synthetic angiograms as well as vessel segmentation masks, leading to learning semantic information about vessels more effectively. Although the diffusion module training is combined with other loss functions, the inference is not iterative but only done in one step, which makes it faster and unique compared to the existing diffusion models. Using various medical vessel datasets, we verify that our model is much superior to existing un-/self-supervised learning methods. Moreover, thanks to the diffusion module, our model is robust to image diversity and noise, suggesting that our model can be an important platform for designing a general vessel segmentation model.

REPRODUCIBILITY

Source code is available at `https://github.com/bispl-kaist/DARL`.

ACKNOWLEDGMENTS

This work was supported in part by the National Research Foundation of Korea under Grant NRF-2020R1A2B5B03001980, in part by Institute of Information & communications Technology Planning & Evaluation (IITP) grant funded by the Korea government(MSIT, Ministry of Science and ICT) (No. 2022-0-00984, Development of Artificial Intelligence Technology for Personalized Plug-and-Play Explanation and Verification of Explanation), in part by the MSIT(Ministry of Science and ICT), under the ITRC(Information Technology Research Center) support program(IITP-2022-2020-0-01461) supervised by the IITP, and in part by the KAIST Key Research Institute (Interdisciplinary Research Group) Project.

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

## A DETAILS OF NETWORK ARCHITECTURE

In this section, we provide details of the generator $G$ proposed in our diffusion adversarial representation learning (DARL) model, which is composed of the diffusion module and the generation module. For the diffusion module, we adapt the network architecture of DDPM (Ho et al., 2020) that has U-Net (Ronneberger et al., 2015) structure, as described in Table 4. The generation module is composed of four consecutive residual blocks (He et al., 2016) with switchable spatially-adaptive denormalization (SPADE) layers, as described in Table 5.

Table 4: Detailed network architecture of the diffusion module. For each block (blk), $C_{i,j}$ is the convolution layer with $i \times i$ kernel and stride length of $j$, $RS_i$ pairs are entry points for residual shortcut path within a block unit, RB is the residual block module, and SA is the self-attention module. GN is the group normalization, and Ch indicates the size of output channel dimension.

| Blk | Diffusion module | | | | | | | | | | | | | | Ch |
|---|---|---|---|---|---|---|---|---|---|---|---|---|---|---|---|
| | Downstream | | | | | | Upstream | | | | | | | | |
| **1** | $C_{3,1}$ | $RS_1$ | RB | $RS_2$ | RB | $RS_3$ | | $RS_3$ | RB | $RS_2$ | RB | $RS_1$ | RB | CB | 1 |
| **2** | $C_{3,2}$ | $RS_1$ | RB | $RS_2$ | RB | $RS_3$ | | $RS_3$ | RB | $RS_2$ | RB | $RS_1$ | RB | UP | 64 |
| **3** | $C_{3,2}$ | $RS_1$ | RB | $RS_2$ | RB | $RS_3$ | | $RS_3$ | RB | $RS_2$ | RB | $RS_1$ | RB | UP | 128 |
| **4** | $C_{3,2}$ | $RS_1$ | RB | $RS_2$ | RB | $RS_3$ | | $RS_3$ | RB | $RS_2$ | RB | $RS_1$ | RB | UP | 128 |
| **5** | $C_{3,2}$ | $RS_1$ | RB SA | $RS_2$ | RB SA | $RS_3$ | | $RS_3$ | RB SA | $RS_2$ | RB SA | $RS_1$ | RB SA | UP | 256 |
| **6** | $C_{3,2}$ | $RS_1$ | RB | $RS_2$ | RB | $RS_3$ | | $RS_3$ | RB | $RS_2$ | RB | $RS_1$ | RB | UP | 256 |
| **Mid** | | | | | RB | | SA | | RB | | | | | | |

*Note*: RB = [$RS_n$ - GN - Swish - $C_{3,1}$ - GN - Swish - $C_{3,1}$ - $RS_n$], SA = [GN - $C_1$ - $C_1$ ],
UP = [Upsample - $C_{3,1}$], CB = [GN - Swich - $C_3$]

Table 5: Detailed network architecture of the generation module. UP is the nearest neighbor upsampling function, $RS_i$ pairs are entry points for residual shortcut path, $C_{i,j}$ is the convolution layer with $i \times i$ kernel and stride length of $j$, and IN is the instance normalization layer. S-SPADE is the proposed switchable SPADE layer that turns on SPADE if the semantic layout is provided, otherwise turns off SPADE and applies IN. Ch indicates the size of output channel dimension.

| Stream | Generation module | | | | | | | | Ch |
|---|---|---|---|---|---|---|---|---|---|
| | Scale | | Conv. | Act. | Norm. | Conv. | Act. | Norm. | |
| **In** | | | $C_7$ | | IN | | ReLU | | 64 |
| **DownBlock1** | | | $C_{3,2}$ | | IN | | ReLU | | 128 |
| **DownBlock2** | | | $C_{3,2}$ | | IN | | ReLU | | 256 |
| **MidResBlock1** | | $RS_1$ | $C_{3,1}$ | ReLU | S-SPADE | $C_{3,1}$ | ReLU | S-SPADE | $RS_1$ | 256 |
| **MidResBlock2** | | $RS_2$ | $C_{3,1}$ | ReLU | S-SPADE | $C_{3,1}$ | ReLU | S-SPADE | $RS_2$ | 256 |
| **UpResBlock1** | UP | $RS_3$ | $C_{3,1}$ | ReLU | S-SPADE | $C_{3,1}$ | ReLU | S-SPADE | $RS_3$ | 128 |
| **UpResBlock2** | UP | $RS_4$ | $C_{3,1}$ | ReLU | S-SPADE | $C_{3,1}$ | ReLU | S-SPADE | $RS_4$ | 64 |
| **Out** | | | $C_7$ | | | | | | 1 |

## B DETAILS OF DATASET

For training the network, as described in Table 6, we utilize the XCAD dataset (Ma et al., 2021) which provides a total of 1,621 unlabeled X-ray coronary angiography frames. We use each first frame that is taken before the contrast agent injection as the real background image. We also generate 1,621 synthetic fractal masks by using the fractal synthetic module proposed by (Ma et al., 2021). The fractal masks are synthesized by drawing rectangles with randomly sampled thickness ranging from 15 to 25 pixels on a black background with a size of 512 x 512. Then, local distortions are taken to each rectangle, including affine transformation with a random scale and rotations with a random angle, resulting in generating masks with various shapes and thicknesses. This reduces the effort to match the real vessel thickness distribution, thus, one can simply synthesize such various fractal masks through the fractal synthetic module. Additional 126 angiography images, along with the ground-truth vessel masks annotated by experienced radiologists, are divided into validation and test sets by 10% and 90%, respectively. We subsample all data into 256×256.

For training the baseline methods, we utilize the same amount of angiography images from the XCAD dataset. In specific, our method and SSVS utilize angiography images, background images,

and synthetic fractal masks for training both segmentation and generation paths, but each data is randomly sampled independently. DA utilizes angiography images and fractal masks for adversarial training. For inferencing DS, we utilize pre-trained network parameters. Redrawing, STEGO and SGC are basically clustering-based methods, which need only angiography images. For all these methods, we use the same $256 \times 256$ images as ours, without any further image processing such as normalization, but augment data through random flipping and 90-degree rotation.

For external test dataset, we utilize two X-ray coronary angiography (XCA) datasets acquired from different machines. 134 XCA dataset is composed of 134 angiography images with the vessel masks labeled by an expert cardiologist (Cervantes-Sanchez et al., 2019). 30 XCA datset is composed of 30 sequences of angiography images (Hao et al., 2020). We utilize one angiography image from each sequence, along with its corresponding ground-truth vessel mask labeled by experts. All the test images are resized to $512 \times 512$. Furthermore, for evaluating cross-organ generalization capability, we utilize retinal imaging datasets. We use DRIVE (Staal et al., 2004) and STARE (Hoover & Goldbaum, 2003) datasets, each of which is composed of 20 retinal images and the corresponding expert-labeled vessel masks. Since retinal imaging is taken under high-resolution, we resize the image into $768 \times 768$ and split into 9 patches with $256 \times 256$.

Table 6: Detailed dataset for training each (A) segmentation and (B) generation path.

| | Input | Train | Validation | Test | | | | |
| --- | --- | --- | --- | --- | --- | --- | --- | --- |
| | | XCAD | XCAD | XCAD | 134 XCA | 30 XCA | DRIVE | STARE |
| (A) Segmentation | Angiogram; $x^a$ | 1,621 | 12 | 114 | 134 | 30 | 20 | 20 |
| | Ground-truth mask | 0 | 12 | 114 | 134 | 30 | 20 | 20 |
| (B) Generation | Background; $x^b$ | 1,621 | - | - | - | - | - | - |
| | Fractal mask; $s^f$ | 1,621 | - | - | - | - | - | - |

## C  ADDITIONAL EXPERIMENTAL RESULTS

### C.1  STUDY ON SYNTHETIC ANGIOGRAM GENERATION

As described in the main paper, a single generation module with the switchable SPADE layers in our model provides both the synthetic angiograms and the vessel segmentation masks by one-step inference, compared to iterative inference steps of other diffusion models. To evaluate the angiogram synthesis performance, we compare our model with the methods of DA and SSVS. These baselines generate the segmentation masks and the synthetic angiograms, but unlike ours, they use two different networks employing the CycleGAN (Zhu et al., 2017) framework. Furthermore, in this study, we adopt an additional baseline method of OASIS (Sushko et al., 2020), one of the SOTA semantic image synthesis models. Under the same condition as ours with an unpaired dataset setting, we train the OASIS model. For all the baseline methods, a total of 1,621 synthetic angiography images are generated using the backgrounds and fractal masks as inputs.

Figure 6 compares the visual results of synthetic angiograms. Compared to the others, our generation module yields the most realistic images that naturally reflect fractal masks on the background and also contain even tiny branches. This verifies that our model maintains consistency on the vessel-like fractal signals, capturing vessel semantic information effectively and leading to the improvement of segmentation performance. Also, we perform the quantitative evaluation on image generation of angiograms and vessel masks using Fréchet inception distance (FID) (Heusel et al., 2017). Table 7 shows that our model achieves much lower FID scores than the comparative methods even though the FID is originally designed for evaluating natural image synthesis, which suggests the superiority of ours in generating angiography images. Also, the proposed DARL model provides the most realistic segmentation masks over the other methods.

### C.2  STUDY ON HYPERPARAMETER SETTING

In the main paper, we report the vessel segmentation results from the model trained with $\alpha = 0.2$ and $\beta = 5$ based on the study of hyperparameter setting, which yields optimal performance in our experiments. To study the effects of hyperparameters on the segmentation performance, using the

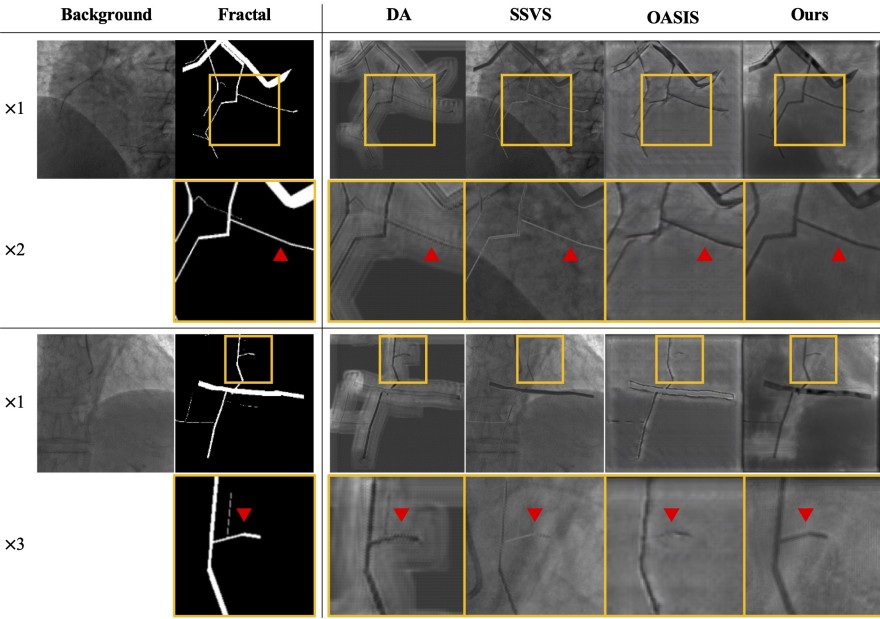

Figure 6: Visual comparison results of the generated fake angiograms. Yellow boxes in ×1 rows are magnified by two or three times in the corresponding bottom rows, respectively. Red triangles indicate remarkable points.

Table 7: Quantitative comparison results using FID score on the image generation of vessel masks and angiograms. Lower FID means that the image generation is more realistic.

| Image generation | DA | SSVS | OASIS | Ours |
|---|---|---|---|---|
| Vessel mask | 123.20 | 149.00 | N/A | 93.49 |
| Angiogram | 261.64 | 191.68 | 307.50 | 177.59 |

proposed loss function, we trained our model with the fixed $\beta = 5.0$ when adjusting the parameter $\alpha$. Similarly, $\alpha$ is fixed with 0.2 when $\beta$ is adjusted. Figure 7 shows graphs of the quantitative evaluation results of IoU, Dice, and Precision metrics according to the hyperparameters of $\alpha$ and $\beta$. We can see that our model can learn the semantic vessel segmentation when the parameter $\alpha$ that controls the adversarial loss $\mathcal{L}_{adv}$ is equal to or more than 0.2, though the performance gradually decreases as $\alpha$ increases. Also, the results show that the highest performance for all metrics is achieved when $\alpha = 0.2$. On the other hand, while our model hardly captures the semantic information of vessels when there is no cycle path in network training (i.e. $\beta = 0$), the model can provide plausible vessel segmentation masks as long as the cycle path exists. Also, when we investigate the segmentation performance according to the $\beta$ that weights the cyclic loss $\mathcal{L}_{cyc}$, the optimal performance is obtained when $\beta = 5.0$.

## C.3 STUDY ON ANGIOGRAM PERTURBATION IN MODEL TRAINING

When training our model, the background images are perturbed by the forward diffusion process with the uniformly sampled time step $t_b \in [0, T]$, whereas the angiograms are perturbed with the time step $t_a \in [0, T_a]$ where $T_a < T$. To investigate the effect of time step size $T_a$ for the angiogram perturbation on model performance, we train our model with different time step sizes by setting $T_a$ as 100, 200, 500, and 1000. Figure 8 shows the quantitative evaluation results. When $T_a$ is less than 200, the vessel segmentation performance on the clean angiograms gets better, but the performance is degraded on the simulated low-dose angiograms corrupted by Gaussian noise with $\sigma$ levels of 25 and 50. Also, when $T_a$ is set over 500, the model shows drastically low performance due to the lack of vascular information from the noisy angiograms. These results imply that the diffusion module can optimally provide latent features including vascular structures as long as the model is trained

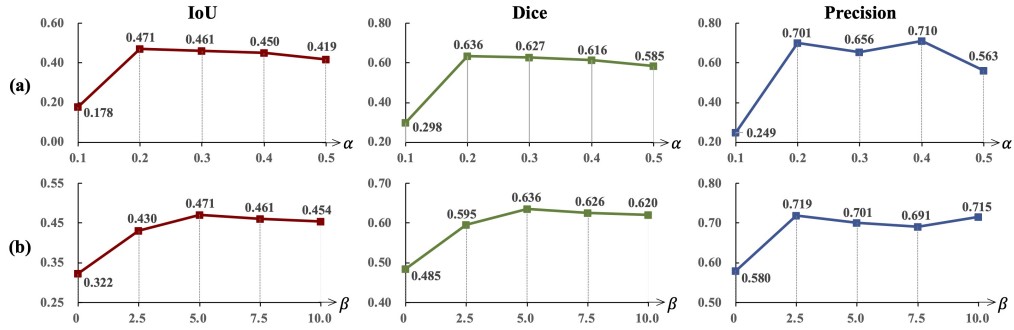

Figure 7: Vessel segmentation performance of our model on the XCAD dataset according to the hyper-parameters of the proposed loss function. Each column shows the average values of quantitative evaluation results with respect to (a) $\alpha$ for the adversarial loss $\mathcal{L}_{adv}$ and (b) $\beta$ for the cyclic reconstruction loss $\mathcal{L}_{cyc}$.

in the setting of $T_a \leq 200$. Also, the proposed implementation makes our DARL robust to noise, which suggests that our model can segment vascular structures even on low-dose medical images.

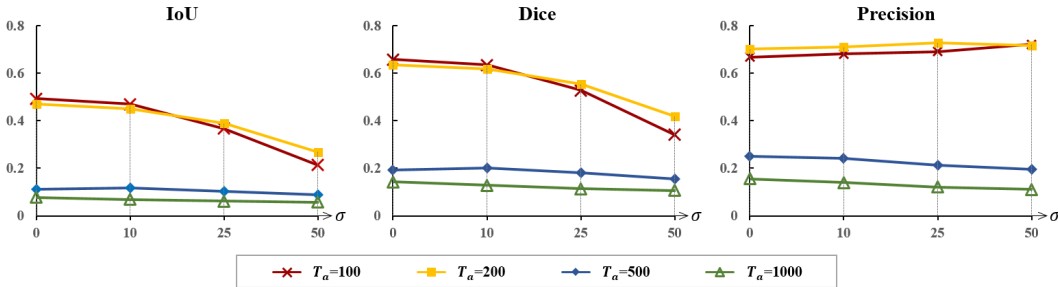

Figure 8: Vessel segmentation performance of our model according to the settings of $T_a$ for the angiogram perturbation. Each graph shows the quantitative evaluation results with respect to the Gaussian noise level $\sigma$.

## C.4 STUDY ON ADVERSARIAL LEARNING WITH TWO DISCRIMINATORS

Our proposed model is trained via adversarial learning with two discriminators $D_s$ and $D_a$ that distinguish real and fake segmentation masks and angiograms, respectively. To confirm that this discriminator setting is optimal, we additionally conduct experiments to train our model without either $D_s$ or $D_a$. As reported in Table 8(a), the model trained without $D_s$ slightly increases the precision but degrades the scores for the IoU and Dice metrics. Also, the model trained without $D_a$ shows inferior segmentation performance as shown in Table 8(b), which may be due to the failure to generate realistic angiograms, making the model relatively hard to learn vessel semantic information. These results suggest that our model with both $D_s$ and $D_a$ is optimal to learn vessel representations.

Table 8: Vessel segmentation performance of our model without the discriminator $D_s/D_a$.

| Method | Discriminators | | Metric | | |
|---|---|---|---|---|---|
| | $D_s$ | $D_a$ | IoU | Dice | Precision |
| Ours | ✓ | ✓ | $\mathbf{0.471}_{\pm \mathbf{0.076}}$ | $\mathbf{0.636}_{\pm \mathbf{0.072}}$ | $0.701_{\pm 0.115}$ |
| (a) | | ✓ | $0.456_{\pm 0.081}$ | $0.622_{\pm 0.078}$ | $\mathbf{0.728}_{\pm \mathbf{0.117}}$ |
| (b) | ✓ | | $0.348_{\pm 0.061}$ | $0.513_{\pm 0.069}$ | $0.496_{\pm 0.106}$ |

### C.5 STUDY ON DIFFUSION MODEL IN LATENT FEATURE ESTIMATION

Recall that the output of the diffusion module is not a simple latent feature of networks but a score function that has spatial information of the data. To show the effect of the diffusion model in our framework, we additionally study our framework by replacing the diffusion module with the autoencoder model. For a fair comparison, We configure the autoencoder by adapting the same DDPM network architecture (Ho et al., 2020) as ours but removing the time embedding vectors and then train without the diffusion loss. Figure 9 shows the latent features and vessel masks of the angiography images. We can observe that while the autoencoder model estimates latent features that include vessels and other similar confusing structures, our proposed framework with the diffusion module represents vessels better in the latent features and provides more accurate segmentation masks. Table 9 also shows that the autoencoder model achieves inferior performance compared to ours. These results indicate that the latent features from the diffusion module allow the generation module to effectively learn vessel representation.

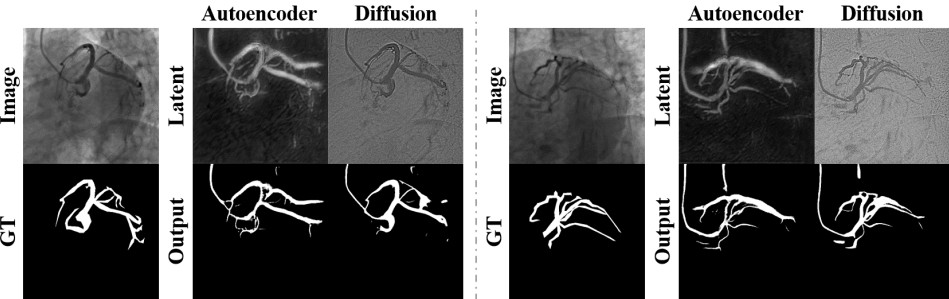

Figure 9: Visual results of vessel segmentation according to the latent feature estimation models.

Table 9: Quantitative evaluation results according to the latent feature estimation models.

| Latent feature estimation | IoU | Dice | Precision |
|---|---|---|---|
| Autoencoder | $0.399_{\pm 0.076}$ | $0.566_{\pm 0.079}$ | $0.621_{\pm 0.109}$ |
| Diffusion (Ours) | $\mathbf{0.471_{\pm 0.076}}$ | $\mathbf{0.636_{\pm 0.072}}$ | $\mathbf{0.701_{\pm 0.115}}$ |

### C.6 STUDY ON MODEL CONTRIBUTION

The contribution of proposed DARL model is further analyzed under supervised, semi-supervised, and self-supervised learning scenarios, and each result on various datasets are described in Table 10.

**Supervised** Firstly, we apply the proposed framework as a supervised model. Here, since there are no labeled pairs in the training dataset, we conduct two supervised segmentation experiments utilizing the 12 pairs of labeled validation data: 1) training our model through only the (A) path, and 2) training our model by giving the labeled data to the (A) path while keeping the (B) path. To evaluate the performance of supervised approaches, we report the results of the best model among models saves at 10 epoch intervals. In Table 10, we can observe that in the few-label scenario, our DARL framework contributes to the segmentation path to achieve superior performance, leading to getting better than the supervised model only with the (A) path. Moreover, it is noteworthy that the performance of our self-supervised model surpasses the supervised approach only using the (A) path by large margins, even though ours is trained without any supervised data.

**Semi-supervised** As the outputs in the (A) and (B) paths of our DARL model can be used as pseudo labels or inputs for vessel segmentation, we train the segmentation model in a semi-supervised manner. In specific, we prepare the segmentation network that is identical network architecture to that of our generation module $G$ for a fair comparison. Then, we train the network by utilizing the (A) path outputs $\hat{s}^v$ as paired pseudo-label for input $x^a$. Similarly, we train the network by using the (B) path outputs $\hat{x}^a$ as paired pseudo-input for label $s^f$. As reported in Table 10, when compared to the model trained using the data from the (A) path, the performance is slightly higher

than ours. This suggests that the generated masks from our model can be used for pseudo-labels for unlabeled data. Also, it is remarkable that although our method estimates the vessel maps and synthetic angiography images simultaneously, our method achieves comparable performance with the semi-supervised method using the pseudo-labels. On the other hand, the model trained using the data from the (B) path shows mostly lower performance than ours, implying the cycle path in our model is more effective to extract vessels.

**Self-supervised** We further test the application of our model to an environment that has no background images. Our model is trained by replacing the background images $x^b$ in the (B) path with the real angiography images $x^a$. Table 10 shows that the segmentation performance of our model without non-contrast background images is comparable or even superior to our model. As the generation module of our DARL takes latent features of input images, our model can synthesize images involving the information of input angiography images, based on semantic masks, and learn vessel features. This can be a unique characteristic compared to other semantic image synthesis models that typically take the images directly.

Table 10: Study on model contribution to various learning frameworks.

| Data | Metric | Supervised (few-label) | | Semi-supervised (pseudo-pair) | | Self-supervised | |
|---|---|---|---|---|---|---|---|
| | | (A) path | (A)+(B) path | Data from (A) | Data from (B) | Ours | Ours w/o BG |
| XCAD | IoU | $0.423_{\pm 0.079}$ | $\mathbf{0.548_{\pm 0.072}}$ | $0.478_{\pm 0.078}$ | $0.445_{\pm 0.072}$ | $0.471_{\pm 0.076}$ | $0.479_{\pm 0.080}$ |
| | Dice | $0.590_{\pm 0.081}$ | $\mathbf{0.705_{\pm 0.062}}$ | $0.643_{\pm 0.073}$ | $0.613_{\pm 0.071}$ | $0.636_{\pm 0.072}$ | $0.644_{\pm 0.076}$ |
| | Precision | $0.637_{\pm 0.134}$ | $\mathbf{0.746_{\pm 0.088}}$ | $0.703_{\pm 0.115}$ | $0.645_{\pm 0.117}$ | $0.701_{\pm 0.115}$ | $0.700_{\pm 0.125}$ |
| **External test: Coronary angiography** | | | | | | | |
| 134 XCA | IoU | $0.273_{\pm 0.161}$ | $0.323_{\pm 0.195}$ | $0.388_{\pm 0.187}$ | $0.378_{\pm 0.179}$ | $\mathbf{0.426_{\pm 0.059}}$ | $0.377_{\pm 0.187}$ |
| | Dice | $0.404_{\pm 0.201}$ | $0.454_{\pm 0.237}$ | $0.531_{\pm 0.211}$ | $0.522_{\pm 0.205}$ | $\mathbf{0.595_{\pm 0.058}}$ | $0.518_{\pm 0.215}$ |
| | Precision | $0.439_{\pm 0.215}$ | $0.453_{\pm 0.236}$ | $0.477_{\pm 0.213}$ | $0.458_{\pm 0.198}$ | $\mathbf{0.781_{\pm 0.118}}$ | $0.469_{\pm 0.208}$ |
| 30 XCA | IoU | $0.365_{\pm 0.042}$ | $\mathbf{0.484_{\pm 0.066}}$ | $0.429_{\pm 0.064}$ | $0.425_{\pm 0.080}$ | $0.427_{\pm 0.184}$ | $0.429_{\pm 0.070}$ |
| | Dice | $0.534_{\pm 0.045}$ | $\mathbf{0.649_{\pm 0.061}}$ | $0.598_{\pm 0.064}$ | $0.592_{\pm 0.082}$ | $0.572_{\pm 0.205}$ | $0.597_{\pm 0.070}$ |
| | Precision | $0.717_{\pm 0.121}$ | $\mathbf{0.829_{\pm 0.083}}$ | $0.772_{\pm 0.124}$ | $0.737_{\pm 0.133}$ | $0.729_{\pm 0.152}$ | $0.762_{\pm 0.124}$ |
| **Cross-modality test: Retinal imaging** | | | | | | | |
| DRIVE | IoU | $0.344_{\pm 0.134}$ | $0.388_{\pm 0.141}$ | $0.365_{\pm 0.148}$ | $0.341_{\pm 0.147}$ | $0.372_{\pm 0.148}$ | $\mathbf{0.392_{\pm 0.135}}$ |
| | Dice | $0.497_{\pm 0.153}$ | $0.544_{\pm 0.153}$ | $0.518_{\pm 0.159}$ | $0.490_{\pm 0.163}$ | $0.525_{\pm 0.161}$ | $\mathbf{0.550_{\pm 0.144}}$ |
| | Precision | $0.617_{\pm 0.249}$ | $\mathbf{0.754_{\pm 0.213}}$ | $0.585_{\pm 0.271}$ | $0.504_{\pm 0.265}$ | $0.617_{\pm 0.271}$ | $0.659_{\pm 0.247}$ |
| STARE | IoU | $0.319_{\pm 0.168}$ | $0.332_{\pm 0.195}$ | $0.375_{\pm 0.187}$ | $0.354_{\pm 0.183}$ | $0.368_{\pm 0.191}$ | $\mathbf{0.393_{\pm 0.183}}$ |
| | Dice | $0.458_{\pm 0.205}$ | $0.464_{\pm 0.235}$ | $0.517_{\pm 0.210}$ | $0.495_{\pm 0.208}$ | $0.508_{\pm 0.216}$ | $\mathbf{0.538_{\pm 0.205}}$ |
| | Precision | $0.529_{\pm 0.267}$ | $\mathbf{0.591_{\pm 0.295}}$ | $0.528_{\pm 0.269}$ | $0.481_{\pm 0.254}$ | $0.537_{\pm 0.280}$ | $0.566_{\pm 0.257}$ |

BG; background images $x^b$

## C.7 STUDY ON IMAGE PROCESSING FOR THE UNSUPERVISED METHODS

For the implementation of comparison methods, we did not perform any image processing except for resizing the images. However, as the performance of unsupervised methods would be affected by the processing such as normalization, we additionally tested the unsupervised methods by applying several normalization methods to the input images. As shown in Figure 10, we processed images using histogram equalization (HE) and contrast limited adaptive histogram equalization (CLAHE). Table 11 shows that the performance of unsupervised methods degrades when using the normalized data. This comes from the angiography images that are hard to visualize only vessel regions due to the confusing background structures even though the CLAHE and HE enhance the image contrast. On the other hand, our method outperforms the other comparative models, suggesting the superiority of our methods.

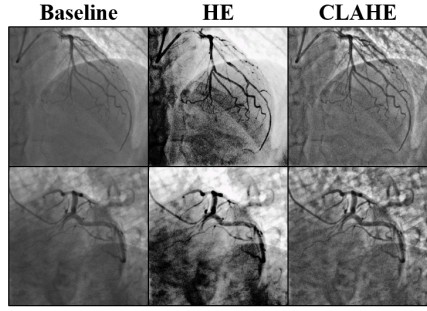

Figure 10: Examples of image processing of HE and CLAHE normalization.

Table 11: The Dice scores for the segmentation performance of unsupervised methods applying HE and CLAHE to the input images.

| Input processing | SGC | Redrawing | DS | Ours |
|---|---|---|---|---|
| Baseline | 0.111 | 0.109 | 0.526 | 0.636 |
| HE | 0.061 | 0.107 | 0.492 | - |
| CLAHE | 0.119 | 0.099 | 0.495 | - |

## C.8 STUDY ON TRAINING COMPLEXITY

Our model has a unified generator module which can perform both the segmentation and the generation tasks simultaneously, by efficiently decreasing training complexity compared to CycleGAN structure of other adversarial learning framework such like DA or SSVS. We estimate floating point operation per second (FLOPS) of each method in Table 12, and prove the cost-effective characteristic of our model.

Table 12: Training complexity (FLOPS).

| Method | (A) path | (B) path | Cycle path | Discriminator | Total |
|---|---|---|---|---|---|
| DA | 121.80 | 121.80 | $121.80 \times 2$ | $6.24 \times 2$ | 499.68 |
| SSVS | 121.80 | 121.80 | $121.80 \times 2$ | $6.24 \times 2$ | 499.68 |
| Ours | 90.66 | 173.51 | $90.66 \times 1$ | $6.24 \times 2$ | 367.31 |

## D ADDITIONAL VISUAL RESULTS OF VESSEL SEGMENTATION

### D.1 RESULTS OF ABLATION STUDY

**Implementation detail** In Section 4.1 of the main paper, we implemented several ablated models of ours. For the methods without the diffusion module (i.e. (a) and (b) in Table 3), the real angiography images are given for the (A) path, and the real background and synthetic fractal masks are given for the (B) path. Here, as there is no diffusion module, the input images are not perturbed. In particular, for the methods without S-SPADE layers in the generation module (i.e. (b) in Table 3), there are two independent generation modules for the (A) and (B) paths, with instance normalization and SPADE as normalization layer for each, so that the input image for each path is given to the corresponding module. On the other hand, for the ablation studies on the loss functions, the data flows are the same as ours.

**Visual results** Figure 11 shows the qualitative comparison results for the ablation studies in the main paper. (a) and (b) show that our model trained without the diffusion module generates the vessel masks including many false positive regions. When compared ours with (c), we can observe that the cycle consistency on the fake vessels allows our model to segment tiny vessels more accurately. Moreover, the comparison of (d) and ours verify that the proposed model achieves better performance, although we solve the segmentation problem with the image generation in that we use L1 loss for the segmentation masks.

### D.2 RESULTS OF OUR DARL MODEL

In this section, we provide additional vessel segmentation results that show the success of our DARL in self-supervised segmentation. Figure 12 shows that our model consistently provides the best performance on XCAD dataset. Also, our model segments vascular structures better than the other baselines even on the external angiography datasets (134 XCA and 30 XCA) and the retinal image datasets (DRIVE and STARE). These results suggest that the proposed model can be used as a general vessel segmentation model for various vascular images. Also, Figure 13 shows the vessel segmentation results on the XCAD data that are corrupted by Gaussian noise with different levels of $\sigma$. The visual results demonstrate that our DARL is the only method which endures harsh corruption and outperforms the baseline methods.

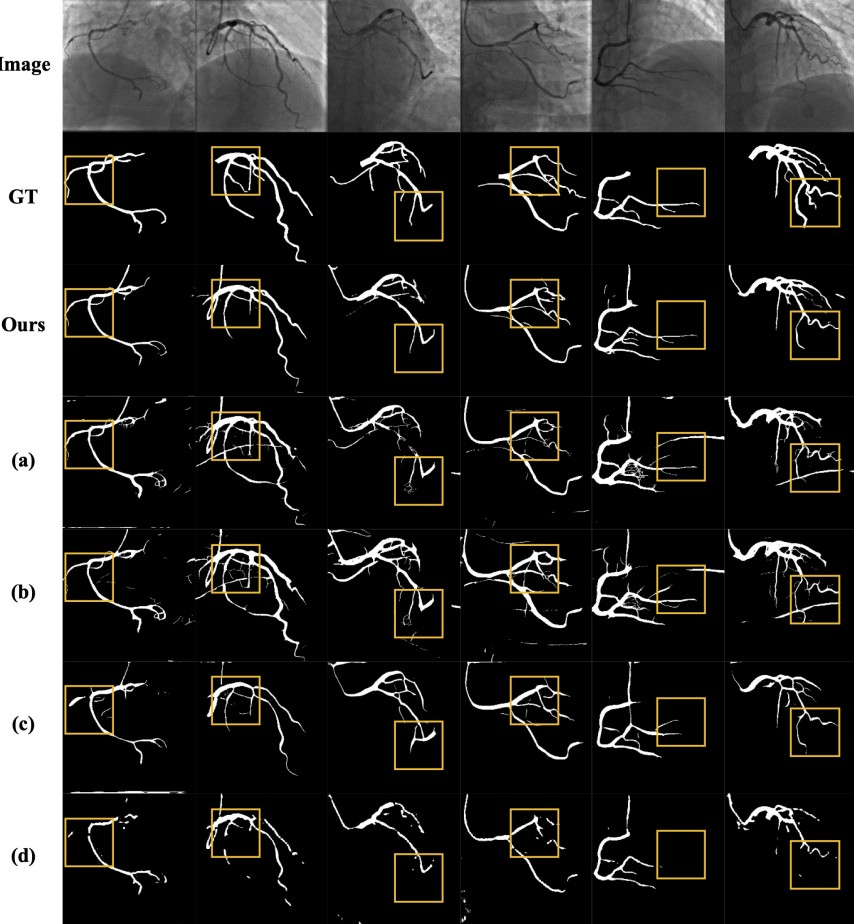

Figure 11: Visual comparison results of the ablation study. (a-d) correspond to each case study (a-d) in Table 3 of the main paper. Yellow boxes denote remarkable parts.

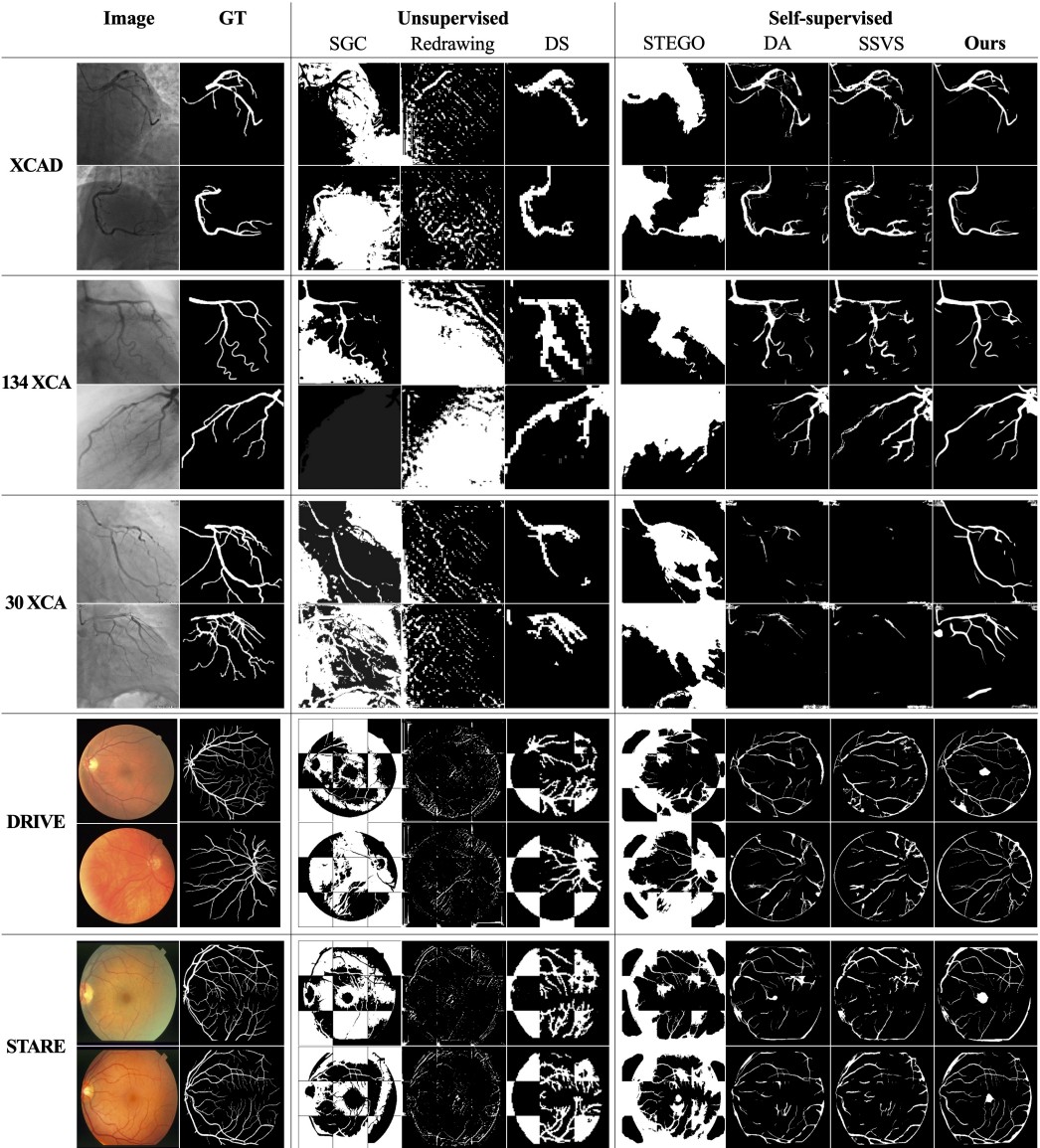

Figure 12: Additional visual comparison results on different angiogram and retinal datasets.

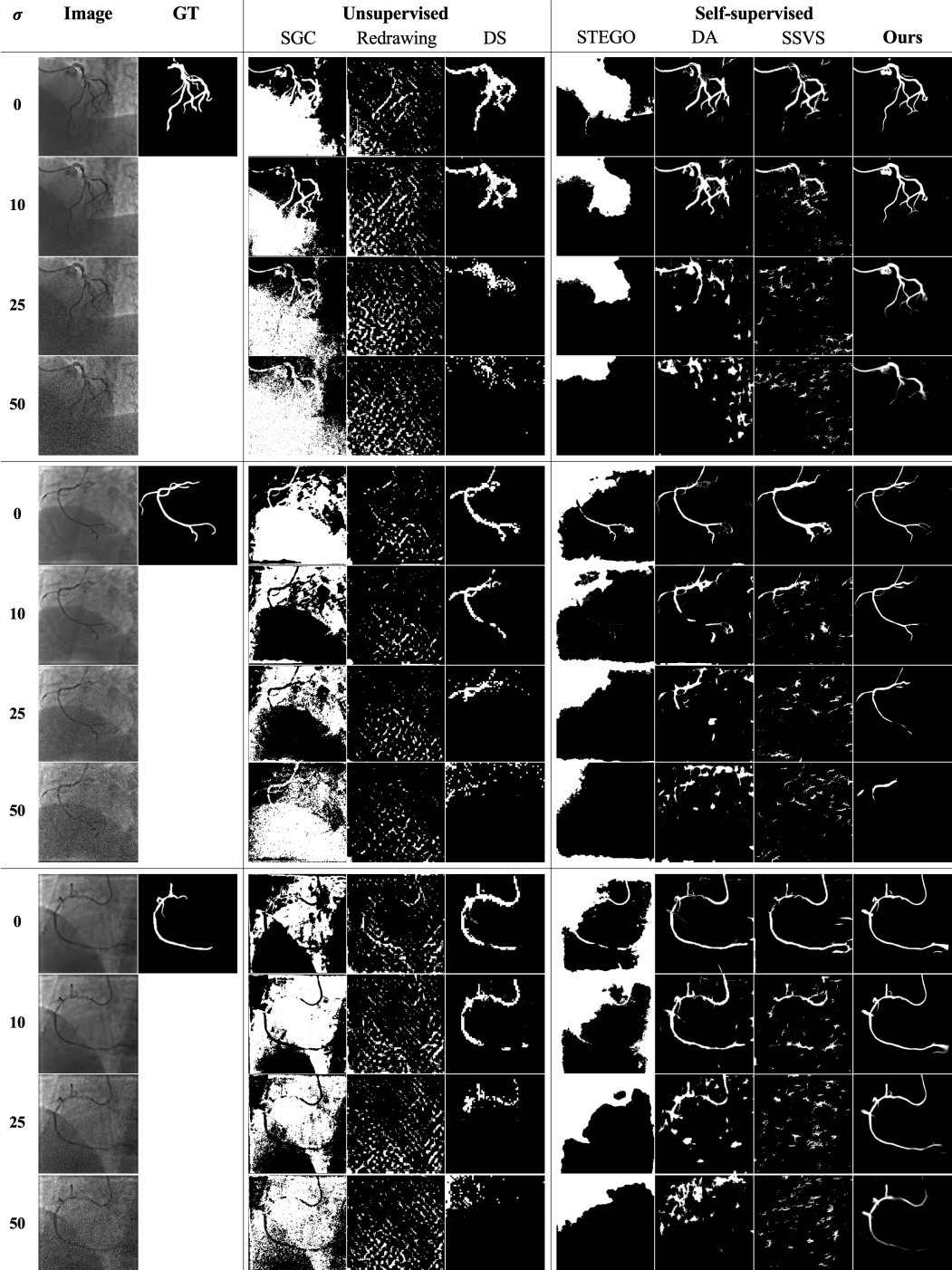

Figure 13: Additional visual comparison results on XCAD datasets with different levels ($\sigma$) of Gaussian noise.

