# OpenReview forum: "Diffusion Adversarial Representation Learning for Self-supervised Vessel Segmentation"
_ICLR.cc/2023/Conference — ICLR 2023 poster_

### Official Review · Reviewer_yNNg · 2022-10-24

**Confidence:** 4
**Clarity, Quality, Novelty And Reproducibility:** the originality is there, but novelty…
**Correctness:** 2
**Technical Novelty And Significance:** 3
**Empirical Novelty And Significance:** 3
**Recommendation:** 6

**Strength And Weaknesses:**

Strengths:
--proposed an effective vessel segmentation method composed of diffusion and adversarial training.
--Obtained a significant improvement of vessel segmentation both on X-ray coronary angiography and fundus images.


Weaknesses:
--the innovation is limited. Diffusion and adversarial are the common approaches, but the authors have not claimed why they can together? In addition, the main dear of this work is cycle reconstruction. It is also common.

--the motivation and writing are unclear. The authors claim that diffusion is employed to learn background image distribution, but the relationship between background distribution and vessel representation is not presented clearly.

--The details of the model and implements are missed. Such as how to train path A and B simultaneously? How to switch SPADE?

--The method may be hard to implement

--It seems that path A of x^a and path B x^b are trained independently. How to make the training consistent?

--In the experiments, the fundus image is introduced. But for training, how to obtain the real noisy background image?

**Summary Of The Paper:**

This paper proposes a self-supervised vessel segmentation method with denoising diffusion model and adversarial training. The proposed method separates the image background and vessel by learning the representation of background images with diffusion model. The main contribution is the combination of diffusion model and generation model for self-supervised vessel segmentation.

**Summary Of The Review:**

the proposed method seems effective and interesting, but the writing is unclear, and some details are missed.

---

> ### Author Response · Authors · 2022-11-13
> **Response to Reviewer yNNg**
>
> >**W1: the innovation is limited. Diffusion and adversarial are the common approaches, but the authors have not claimed why they can together? In addition, the main dear of this work is cycle reconstruction. It is also common.**
> - Thanks for your constructive comment. While the diffusion models have shown impressive performance in learning image representations and generating diverse images, the sampling process takes a relatively long time. Regarding this, we would like to emphasize that we propose a novel non-iterative diffusion model by adapting the adversarial learning model. Specifically, based on the observation that the diffusion model estimates the noise added to the perturbed input data, and the adversarial learning model generates images for given the noisy vectors, we can naturally connect the diffusion model with the adversarial model. This allows our model not only to generate images in real time but also to segment vessels with robustness to noises and various modalities. We have described the motivation of our model in detail in Section 1.
> - Also, although we agree that cycle reconstruction is common when handling unpaired datasets, our contribution is mainly in the use of the diffusion model for self-supervised segmentation. When training our DARL model, with the help of the cycle path, our model can extract vessels more effectively, as demonstrated in Table 3 of Section 4.
>
> >**W2: the motivation and writing are unclear. The authors claim that diffusion is employed to learn background image distribution, but the relationship between background distribution and vessel representation is not presented clearly.**
> - Thanks for your comment. We have clarified the description of vessel representation from the diffusion module in Section 1. As the diffusion module is designed to intensively learn the background signal, allowing the module to regard the vessels in angiography images as outlier when estimating latent features. Accordingly, the vessel structures can be represented in the output of the diffusion module, which guides the generation module to effectively segment the vessels.
>
> >**W3: The details of the model and implements are missed. Such as how to train path A and B simultaneously? How to switch SPADE?**
> - We have clarified the generation module for the (A) and (B) paths in Section 3. Also, the reviewer is kindly reminded that we have already described the switchable SPADE layer and the training detail of our framework. Specifically, the (A) and (B) paths of our model can be implemented simultaneously by using the proposed generation module with switchable SPADE layers. Here, the learnable parameters except for the normalization layer of the generation module are shared on both two paths. When the angiography images are given to the model in the (A) path, the instance normalization layer is used. In contrast, when the background images and synthetic fractal masks are given to the model in the (B) path, the SPADE layer is utilized. As described in Section 3.1.1 the loss functions for each path are computed and the model is updated by minimizing the entire loss.
>
> >**W4: The method may be hard to implement.**
> - We would like to kindly assure the reviewer that we provided the source code to implement our method in Supplementary material. We will release the code and share the link in the final version.
>
> >**W5: It seems that path A of $x^a$ and path B $x^b$ are trained independently. How to make the training consistent?**
> - We would like to kindly remind the reviewer that our method presents a single model even though there are two paths of (A) and (B) for angiography images and background images, respectively. As in the response to the above W3, the learnable parameters of the generation module are shared in those two paths and the model is trained at once.
>
> >**W6: In the experiments, the fundus image is introduced. But for training, how to obtain the real noisy background image?**
> - The reviewer is kindly reminded that the fundus images of the DRIVE and STARE datasets are used to verify the cross-organ generalization capability of our model. Thus, we did not train our model on the fundus images but tested the generalization using the model only trained on the XCAD dataset. We have clarified this in Section 4.1.

---

> ### Author Response · Authors · 2022-11-22
> **We are looking forward to your feedback.**
>
> Dear Reviewer yNNg,
>
> We sincerely appreciate your time and efforts in reviewing our paper, and your constructive comments. We would like to kindly remind you that we tried our best to respond to your concerns with additional experiments, etc. Could you please go over our responses and let us know if there are any remaining issues?
>
> Best regards,
>
> Authors

---

### Official Review · Reviewer_xqnu · 2022-10-24

**Confidence:** 4
**Correctness:** 3
**Technical Novelty And Significance:** 3
**Empirical Novelty And Significance:** 3
**Recommendation:** 6

**Clarity, Quality, Novelty And Reproducibility:**

The proposed DARL framework is generally clearly described, and the network architecture specified in the appendix. The incorporation of diffusion for latent feature estimation in this context appears largely novel. Network architecture details are provided in the appendix, and the datasets used are public.


**Strength And Weaknesses:**

Strengths:

1. Novel application of diffusion towards task-and-noise robust vessel segmentation.

2. Ablation experiments performed to quantify contributions of various modules/losses.

3. Extensive comparisons against other self-supervised and unsupervised methods.

4. Additional experiments described in appendix.

Possible Weaknesses/Considerations:

5. The nature of the background angiography images might be explained further, as readers may not be familiar with the particular task. Are these background images obtained in the same way as the vessel images, just from areas without blood vessels? If so, should their background distribution be expected to be similar to the areas with blood vessels?

6. Some of the most obvious/natural comparisons might not have been attempted. In particular, the inclusion of the diffusion module appears for the estimation of latent features. However, such an estimation of latent features appears doable with an (auto)encoder, possibly with fully-connected layers for both the encoder and decoder within. The support for the DDPM diffusion model for estimating the latent features thus might not be fully established.

7. Similar to the above, the quality of the synthetic angiography/vessel segmentation images produced by the generation module does not appear benchmarked against state-of-the-art GANs for common metrics such as Inception score etc.

8. Moreover, once synthetic angiography images are generated from corresponding (fractal) vessel masks in Path B, the image-vessel mask pairs should constitute acceptable training data for common image segmentation models (e.g. U-Net and variants; i.e. Type B Ground Truth). The same applies to the estimated vessel masks from Path A (i.e. Type A Ground Truth). The specific contribution of the proposed DARL framework might thus be clarified.

9. For the experiments, it might be clarified as to whether any (global) image preprocessing (e.g. CLAHE) was applied. Unsupervised methods would appear particularly disadvantaged without such normalization.

10. The quality of the vessel segmentations appears possibly sensitive to the fractal masks as generated by the fractal synthetic module (from Ma et al., 2021). In particular, the correlation between the (distribution of) thicknesses of the fractal vessels and the actual vessels in the synthetic angiography images, appears critical towards achieving good IoU scores. Was there any attempt to match true vessel thickness distributions, or to optimize the predicted vessel thicknesses against ground truth?

11. For the ablation study in Section 4, it might be explained in greater detail how the model works without the diffusion module, i.e. what the inputs/data flow is then like, possibly in the appendix.


**Summary Of The Paper:**

This paper introduces a diffusion adversarial representation learning (DARL) model, for self-supervised vessel segmentation. The major novelty appears to be the usage of a diffusion module for estimating latent features, which are then used by a generation model to estimate both vessel segmentation masks and synthetic angiograms, through switchable spatially-adaptive denormalization (SPADE) layers.

The estimated output depends on the input, which admits two paths: Path A accepts a real, noisy angiography image, and estimates the vessel segmentation mask (the main task). Path B accepts a real noisy background image and a vessel-like fractal mask, to generate a synthetic angiography image that attempts to incorporate vessels corresponding to the input fractal mask. The synthetic angiography images can then be forwarded through Path A to apply cycle consistency within the LSGAN framework. Experiments are done on the public XCAD datasets, and external XCA datasets, and also applied to out-of-domain retinal imaging datasets. DARL was shown to outperform both unsupervised baseline methods, and recent self-supervised methods such as STEGO, DA and SSVS.


**Summary Of The Review:**

The proposed DARL framework introduces a diffusion module towards self-supervised vessel segmentation exploiting cycle consistency. Segmentation results appear superior to other recent approaches, and robust across input noise and specific task.

---

> ### Author Response · Authors · 2022-11-13
> **Response to Reviewer xqnu (4/4)**
>
> >**W5: For the experiments, it might be clarified as to whether any (global) image preprocessing (e.g. CLAHE) was applied. Unsupervised methods would appear particularly disadvantaged without such normalization.**
> - Thanks for your constructive comment. We have clarified the input image processing of dataset in the Appendix B. Also, per the reviewer’s suggestion that the performance of unsupervised methods would be affected by the input normalization, in Appendix C.7, we have reported the results of the comparisons using the pre-processed angiography images through simple histogram equalization (HE) and contrast limited adaptive histogram equalization (CLAHE). As shown in Resp_Table 8, the performance of comparative methods degrades when using the normalized data. This comes from the angiography images that are hard to visualize only vessel regions due to the confusing background structures even though the CLAHE and HE enhance the image contrast.  On the other hand, our method outperforms the baseline models, suggesting the superiority of our methods.
>
> ***Resp_Table 8.*** The Dice scores for the segmentation performance of unsupervised methods applying HE and CLAHE to the input images.
> |     Input processing       |   SGC  |  Redrawing  |  DS  | Ours |
> |:---------------:|:-----:|:------:|:-----:|:-----:|
> |    Baseline  | 0.111 | 0.109 | 0.526 | 0.636 |
> |   HE             | 0.061 | 0.107 |  0.492  |   -   |
> |   CLAHE        | 0.119 | 0.099 |  0.495  |   -   |
>
> >**W6: The quality of the vessel segmentations appears possibly sensitive to the fractal masks as generated by the fractal synthetic module (from Ma et al., 2021). In particular, the correlation between the (distribution of) thicknesses of the fractal vessels and the actual vessels in the synthetic angiography images, appears critical towards achieving good IoU scores. Was there any attempt to match true vessel thickness distributions, or to optimize the predicted vessel thicknesses against ground truth?**
> - As commented by the reviewer, we synthesized the fractal masks by using the fractal synthetic module by Ma et al., 2021 whose source code is publicly available with suggested parameters. According to the work of Ma et al., 2021, fractal masks are synthesized by drawing rectangles with randomly sampled thicknesses ranging from 15 to 25 pixels on a black background with a size of 512 x 512. Then, local distortions are taken to each rectangle, including affine transformation with a random scale and rotations with a random angle, resulting in generating masks with various shapes and thicknesses. This reduces the effort to match the real vessel thickness distribution, and thus, one can simply synthesize such various fractal masks through the fractal synthetic module. We have added this detail in Appendix B.
>
> >**W7: For the ablation study in Section 4, it might be explained in greater detail how the model works without the diffusion module, i.e. what the inputs/data flow is then like, possibly in the appendix.**
> - Thanks for your constructive comment. For the method without the diffusion module, the model generates desired outputs using the real images, not the latent features of the score function. In other words, the real angiography images are given for the (A) path, and the real background and synthetic fractal masks are given for the (B) path. We have described the detail of the ablation study in the Appendix D.1 of the revised paper.

---

> ### Author Response · Authors · 2022-11-13
> **Response to Reviewer xqnu (3/4)**
>
> >**W4: Moreover, once synthetic angiography images are generated from corresponding (fractal) vessel masks in Path B, the image-vessel mask pairs should constitute acceptable training data for common image segmentation models (e.g. U-Net and variants; i.e. Type B Ground Truth). The same applies to the estimated vessel masks from Path A (i.e. Type A Ground Truth). The specific contribution of the proposed DARL framework might thus be clarified.**
> - Thanks for your valuable comment. We have clarified the contribution of our work in Section 1. First of all, we would like to kindly assure the reviewer that we proposed a self-supervised segmentation model that learns vessel representations without the ground-truth labeled data for the real angiography images. Also, the reviewer is kindly reminded that we have already used the synthetic angiography images from the (B) path as new training data in our framework.
> - To compare our model over the segmentation models using pseudo-pair generated in the (A) or (B) paths, we also conducted an experiment of semi-supervised learning (See Appendix C.6 and Table 10). As shown in the brief results of Resp_Table 7, when compared to the segmentation model trained using the angiography images and generated masks of the (A) path, the performance is slightly higher than our self-supervised methods. This suggests that the generated masks from our model can be used for pseudo-labels for unlabeled data. Also, it is remarkable that although our method estimates the vessel maps and synthetic angiography images simultaneously, our method achieves comparable performance with the semi-supervised method using the pseudo-labels. On the other hand, the segmentation model trained using the semantic fractal labels and the synthetic angiography outputs of the (B) path shows mostly lower performance than ours, implying the cycle path in our model is more effective to extract vessels.
> - In addition, one of the main contributions of our method is that we design the generation module to perform both image segmentation and angiography generation through the proposed switchable SPADE layers. Accordingly, as each semantic fractal mask in the (B) path can be regarded as the ground-truth label for the generated angiography image, as illustrated in Figure 2(B), we feed the synthetic images in the (A) path again and compute the loss function between the vessel segmentation map and the label. Here, the reason that we design the unified switchable generation module, rather than employ the common image segmentation model after the angiography image generation, is to not only reduce the computational cost but also to yield a synergy effect in learning vessel representation by extracting proper features for angiogram synthesis. Also, since our model takes the perturbed input data, our model allows the segmentation to be robust to the noisy images.
>
> ***Resp_Table 7.*** Experimental results with Dice scores of our method over the semi-supervised learning.
> |  Method | XCAD|  134 XCA  | 30 XCA  | DRIVE | STARE |
> |:------------------------------------:|:-----:|:------:|:-------:|:------:|:------:|
> |    Semi-supervised: Data from (A) path  | 0.643 | 0.531 | 0.598 | 0.518 | 0.517 |
> |    Semi-supervised: Data from (B) path  | 0.613 | 0.522 | 0.592 | 0.490 | 0.495 |
> |    Self-supervised (Ours)  | 0.636 | 0.595 | 0.572 | 0.525 | 0.508 |

---

> ### Author Response · Authors · 2022-11-13
> **Response to Reviewer xqnu (2/4)**
>
> >**W2: Some of the most obvious/natural comparisons might not have been attempted. In particular, the inclusion of the diffusion module appears for the estimation of latent features. However, such an estimation of latent features appears doable with an (auto)encoder, possibly with fully-connected layers for both the encoder and decoder within. The support for the DDPM diffusion model for estimating the latent features thus might not be fully established.**
> - Thanks for your constructive comment. Although we agree that the estimation of latent features can be done with an autoencoder, we would like to emphasize that the output of the diffusion module is not a simple latent feature of networks but a score function that has spatial information of the data. To show the effect of the diffusion model in our framework, we additionally study our framework by replacing the diffusion module with the autoencoder model. For a fair comparison, we configure the autoencoder by adapting the same DDPM network architecture as ours but removing the time embedding vectors, and then train without the diffusion loss. As a result, while the autoencoder model estimates latent features that include vessels and other similar confusing structures, our proposed framework with the diffusion module represents vessels better in the latent features and provides more accurate segmentation masks. (See below Resp_Table 5 and Appendix C.5 in the revised paper). This indicates that the latent features from the diffusion module allow the generation module to effectively learn vessel representation.
>
> ***Resp_Table 5.*** Vessel segmentation results according to the latent feature estimation models.
> |     Model        |   IoU |  Dice  |  Precision |
> |:-----------------:|:--------:|:-------------:|:-----:|
> |    Autoencoder     | 0.399 | 0.566 | 0.621 |
> |    Diffusion   	  | 0.471 | 0.636 | 0.701 |
>
> >**W3: Similar to the above, the quality of the synthetic angiography/vessel segmentation images produced by the generation module does not appear benchmarked against state-of-the-art GANs for common metrics such as Inception score etc.**
> - We would like to kindly remind the reviewer that our proposed model is focused on vessel segmentation rather than image generation. During the training phase, the generated angiography images in our model guide the generation module to learn the vessel features through the cycle path. Accordingly, in our experiments, we evaluated the segmentation performance using the Dice, Precision, and IoU metrics.
> - Nevertheless, per the reviewer’s suggestion, we also computed the Frechet inception distance (FID) for the estimated vessel segmentation maps. As reported in Resp_Table 6, our proposed DARL model provides high-quality segmentation masks over the other segmentation methods. We have added these results in the Appendix C.1 of the revised paper.
> - In addition, we have further analyzed the quality of the generated angiography images from ours. For the comparisons, we implemented several generative models of DA, SSVS, and OASIS [1] with the same condition as ours. Resp_Table 6 shows that our model achieves much lower FID scores than the comparative methods even though the FID is originally designed for evaluating natural image synthesis, which suggests the superiority of the proposed method in generating angiography images as well as segmentation maps. We also added this analysis in the Appendix C.1 of the revised paper.
>
> ***Resp_Table 6.*** Quantitative comparison results using FID score on the image generation of vessel masks and angiograms. Lower FID means that the image generation is more realistic.
> |                    | DA  |  SSVS  |  OASIS  | Ours |
> |:--------------------:|:------:|:-------:|:------:|:------:|
> |  Segmentation  | 123.20 | 149.00 | N/A |  **93.49** |
> |  Generation        | 261.64 | 191.68 | 307.50 | **177.59** |
>
> [1] Sushko, Vadim, et al. OASIS: Only Adversarial Supervision for Semantic Image Synthesis. International Journal of Computer Vision (2022): 1-21.

---

> ### Author Response · Authors · 2022-11-13
> **Response to Reviewer xqnu (1/4)**
>
> >**W1: The nature of the background angiography images might be explained further, as readers may not be familiar with the particular task. Are these background images obtained in the same way as the vessel images, just from areas without blood vessels? If so, should their background distribution be expected to be similar to the areas with blood vessels?**
> - Thanks for your comment. Per the reviewer’s suggestion, we have described the angiography and background images in detail in Section 3. Specifically, note that the blood vessels are hardly visualized in X-ray images since the attenuation coefficient of the vessels is similar to the adjacent soft tissues. To improve the visualization of the blood vessels, contrast agents are injected into blood vessels, which allows scanning of the contrast-enhanced images called “angiography” images. In contrast, the "background" images in this paper refer to the non-contrast images scanned before the contrast agent is injected. Therefore, although background images and angiography images are obtained in the same way, as they are taken at different times, the image contrasts are different and the images are not aligned due to the movements of patients.

---

> ### Author Response · Authors · 2022-11-22
> **We are looking forward to your feedback.**
>
> Dear Reviewer xqnu,
>
> We sincerely appreciate your time and efforts in reviewing our paper, and your constructive comments. We would like to kindly remind you that we tried our best to respond to your concerns with additional experiments, etc. Could you please go over our responses and let us know if there are any remaining issues?
>
> Best regards,
>
> Authors

---

### Official Review · Reviewer_qGAg · 2022-10-24

**Confidence:** 4
**Correctness:** 4
**Technical Novelty And Significance:** 4
**Empirical Novelty And Significance:** 4
**Recommendation:** 6

**Clarity, Quality, Novelty And Reproducibility:**

- I don't find the writing particularly clear and had to read it a few times to understand the details. While this may just be a reflection of this reviewer's intellect, I would appreciate attempts at making the meaning clearer.

- The quality could be improved a bit by clarifying experimental details (see above) and further discussion of limitations and the relevance of the results as requested above.

- As far as I am aware this is novel work and I am interested in trying it out myself.

- I am not sure the experiments are easy to reproduce. There is no mention of code repository and given my difficulties in understanding the work, I am not convinced I could reproduce it exactly.


**Strength And Weaknesses:**

Strengths
---------
- Addresses the important and relevant problem of vessel segmentation with limited supervision.

- The approach makes sense and seems to show convincing results.

- Novel use of diffusion models for self-supervised learning.

- Results appear to be a good deal better than state of the art.


Weaknesses
----------
- The method appears to rely on both contrast and non-contrast images and this could limit the usability of the approach as not all usecases have both. Perhaps the authors could clarify the importance of this requirement and limitation in the paper?

- I find it unclear what amounts of labelled and unlabelled data is used for the self-supervised experiments. Could the authors clarify this? Unsupervised and self-supervised learning is mostly relevant if it can be used to achieve clinically relevant results with fewer labels. The results reported on the DRIVE and STARE datasets do not seem particularly convincing compared to what has been reported by supervised techniques. I would find it relevant to discuss this and compare to fully supervised methods.

- Another potential problem with relying on both contrast and non-contrast images is that differences between the modalities are actually a quite large source of signal for these problems. Of course one can claim that this source of information is in some sense free from supervision, but to make the comparison with the other methods more fair, perhaps it would make sense to comment on the degree the other methods can actually exploit this information as well.

- The increasing training complexity of the two adversarial networks is used as an argument against Ma et al. in the Introduction. Is there any concrete evidence of this being a drawback in comparison to the proposed work? What is the training complexity of these approaches? For comments like that to be supported, I would expect measurements of FLOPS or similar.

- Figure 2 legend do not describe the content with enough detail. What are all the variables, subscripts and superscripts?

- "Experimental results show that our method outperforms existing unsupervised and self-supervised learning methods in the absence of labeled data for training.". I find a good argument for including exactly these methods as comparison methods to be lacking. Could the authors justify why these methods are a good representation of the state of the art?

- "most unsupervised methods fail to endure drastic performance degradation when they are applied to vessel segmentation tasks.", are you sure this is what you mean to say?

- No code repository makes it hard to reproduce.


**Summary Of The Paper:**

The paper proposes a self-supervised technique for vessel segmentation based on a combination of a denoising diffusion probabilistic model and adversarial learning. The network uses contrast and non-constrast images of vessels and uses the non-contrast images to learn a distribution of the background that is then used to identify the foreground vessels. The approach is evaluated for vessel segmentation in X-ray coronary angiography using two datasets and for retinal vessel segmentation using the DRIVE and STARE datasets in comparison to 6 other recent methods.

**Summary Of The Review:**

I have put this as marginally above the acceptance threshold based on the above reasons, but I think the manuscript could be strengthened quite a bit by going over the writing.

---

> ### Author Response · Authors · 2022-11-13
> **Response to Reviewer qGAg (3/3)**
>
> >**W4: The increasing training complexity of the two adversarial networks is used as an argument against Ma et al. in the Introduction. Is there any concrete evidence of this being a drawback in comparison to the proposed work? What is the training complexity of these approaches? For comments like that to be supported, I would expect measurements of FLOPS or similar.**
> - Thanks for your valuable comment. Per the reviewer's suggestion, we measured FLOPS for each component of the networks in our model and several comparison methods. As described in the below Resp_Table 4, by utilizing around 70% of total FLOPS to that of SSVS (Ma et al., 2021), our method can efficiently perform both the segmentation and the generation tasks with superior performance. Compared to the DA and SSVS that have CycleGAN structures, our model processes just one cycle path and utilizes a relatively lightweight architecture, which makes the proposed method cost-effective for training. Moreover, when testing the model, as we only utilize the (A) path, the lightweight structure has advantages in fast inference and memory consumption. We have discussed the model complexity in Appendix C.8 of the revised paper.
>
> ***Resp_Table 4.*** Study on training complexity (FLOPS)
> | Method | (A) path | (B) path | Cycle path | Discriminator | Total |
> |:-------:|:-------:|:-------:|:-----------:|:---------:|:-------:|
> |  DA    | 121.80 | 121.80 | 121.80 x 2 | 6.24 x 2 |  499.68 |
> |  SSVS  | 121.80 | 121.80 | 121.80 x 2 | 6.24 x 2 |  499.68 |
> |  Ours   | 90.66 | 173.51 | 90.66  | 6.24 x 2 | **367.32** |
>
> >**W5: Figure 2 legend does not describe the content with enough detail. What are all the variables, subscripts and superscripts?**
> - Thanks for the careful comment. We have added detailed descriptions of the contents in Figure 2 in the revised paper.
>
>
> >**W6: "Experimental results show that our method outperforms existing unsupervised and self-supervised learning methods in the absence of labeled data for training.". I find a good argument for including exactly these methods as comparison methods to be lacking. Could the authors justify why these methods are a good representation of the state of the art?**
> - Thanks for your careful comment. We have narrowed down the claim and revised it as follows:
> “Experimental results show that our method outperforms several baseline methods by large margins for vessel segmentation tasks in the absence of labeled data.”
> - Also, we would like to kindly remind the reviewer that the outperformance of our method in vessel segmentation is contributed from both the diffusion module and the generation module with switchable SPADE layers, which is verified by several experiments in Section 4.1. Specifically, the diffusion module of our model learns a score function that has spatial information, and the following switchable generation module uses it as a latent feature for capturing better representations for both generation and segmentation tasks. It is demonstrated in the ablation study on the diffusion module in Table 3(a). Also, the switchable SPADE layers allow our model to integrate the segmentation and generation tasks in a unified framework. With the ablation study that separates the unified framework into two networks (i.e. Table 3(b)), we have confirmed our remarkable performance gain, suggesting that our model can boost synergies in learning representations between the two different tasks.
>
> >**W7: "most unsupervised methods fail to endure drastic performance degradation when they are applied to vessel segmentation tasks.", are you sure this is what you mean to say?**
> - Thanks for your careful observations. To avoid the potential confusion, we have revised the sentence as follows:
> “..., many unsupervised methods designed for natural image segmentation may get degraded performance when they are applied to vessel segmentation of noisy angiography images.”
>
> >**W8: No code repository makes it hard to reproduce.**
> - We would be happy to disclose the source code repository in the final version, and we have already submitted the source code as supplementary material to ensure anonymity.

---

> > ### Comment · Reviewer_qGAg · 2022-11-22
> > **Thanks**
> >
> > Addition of FLOPS to the table goes some of the way to support the comment. It is a bit unclear if this is for a single pass or if it includes potential differences in convergence rates.
> >
> > Inclusion of code is great.

---

> > > ### Author Response · Authors · 2022-11-23
> > > **Thanks for your feedback.**
> > >
> > > Thanks for your feedback and constructive comments. The reported FLOPS is for a single pass for each method. We fixed total training epochs for each method to avoid potential differences in convergence rates. We will clarify the description of computing FLOPS in the final version.

---

> ### Author Response · Authors · 2022-11-13
> **Response to Reviewer qGAg (2/3)**
>
> >**W2-1: I find it unclear what amounts of labeled and unlabeled data is used for the self-supervised experiments. Could the authors clarify this? Unsupervised and self-supervised learning is mostly relevant if it can be used to achieve clinically relevant results with fewer labels.**
> - Thanks for the valuable comment. We would like to kindly assure the reviewer that our method is trained in an unsupervised manner so that we do not utilize any labeled data from the XCAD dataset. In Table 6 of Appendix B in the revised paper, we have provided a detailed explanation of the number of data we used for each path when training our model.
>
> >**W2-2: The results reported on the DRIVE and STARE datasets do not seem particularly convincing compared to what has been reported by supervised techniques. I would find it relevant to discuss this and compare it to fully supervised methods.**
> - The reviewer is kindly reminded that the use of the DRIVE and STARE retinal imaging datasets in testing aims to demonstrate the cross-modal generalizability of our self-supervised method. Even so, we compared our cross-modal performance to several SOTA methods of Zhou, et al. [1] and Sharif Amit, et al. [2] that are trained in a fully supervised manner using the DRIVE or STARE datasets. As expected, in Resp_Table 3, the cross-modality performance is inferior to that of the SOTA methods. This may be because the retinal vessel structure is composed of tiny capillaries and the retinal image has RGB channels, which is different from the grayscale angiography image on which our method is trained.
> - That said,  the reviewer is kindly reminded that we also conducted a supervised segmentation experiment by training our model through only the (A) path using the 12 pairs of labeled validation data (See the paragraph of Supervised of Appendix C.6 and Table 10). In Resp_Table 3, it is noteworthy that the performance of our self-supervised model still surpasses the supervised approach only using the (A) path, even though ours is trained without any supervised data.
>
> ***Resp_Table 3.*** Study on cross-modality validation on retinal imaging dataset (Dice for segmentation performance).
> |           Model               | Train set | Test type | DRIVE | STARE |
> |-----------------------------|:--------:|:-----------:|-------:|-------:|
> | Self-supervised: Ours      |  XCAD   | Cross-modal | 0.525  | 0.508 |
> | Supervised: Ours w/ only (A) Path      |  XCAD   | Cross-modal | 0.497 | 0.458 |
> | Supervised: Y, Zhou, et al. [1]        |   DRIVE  |  Internal 	| 0.832  |    -    |
> | Supervised: K, Sharif Amit, et al. [2]|   DRIVE & STARE |  Internal | 0.870  |    0.837    |
>
> [1] Yuqian, Zhou, et al. Study Group Learning: Improving Retinal Vessel Segmentation Trained with Noisy Labels. In International Conference on Medical Image Computing and Computer-Assisted Intervention, pp. 57–67, 2021.
>
> [2] Kamran, Sharif Amit, et al. V-GAN: segmenting retinal vascular structure in fundus photographs using a novel multi-scale generative adversarial network. In International Conference on Medical Image Computing and Computer-Assisted Intervention, pp. 34–44, 2021.
>
> >**W3: Another potential problem with relying on both contrast and non-contrast images is that differences between the modalities are actually a quite large source of signal for these problems. Of course one can claim that this source of information is in some sense free from supervision, but to make the comparison with the other methods more fair, perhaps it would make sense to comment on the degree the other methods can actually exploit this information as well.**
> - We appreciate your constructive point. Among the baseline methods, SSVS utilizes both contrast and non-contrast images for training. Other unsupervised baseline methods are mostly based on clustering techniques, which need no non-contrast images nor synthetic fractal masks for training the network. Therefore, to disclose the unbalanced dataset usage, we have described the data information for each method in Appendix B of the revised paper.
> - Additionally, as described in the Self-supervised column of Table 10 in the revised paper, we also verify the usability of our model in an environment of datasets without non-contrast background images, by achieving comparable or even superior to the proposed model. Therefore, we would like to kindly assure the reviewer that the non-contrast information is not an essential factor for our model to achieve the SOTA performance.

---

> > ### Comment · Reviewer_qGAg · 2022-11-22
> > **Thank you**
> >
> > I appreciate the very extensive reply with new experimental data. I believe it is helpful to see the results in comparison to supervised methods. It is also good to know the degree to which the methods rely on both contrast and non-contrast.

---

> ### Author Response · Authors · 2022-11-13
> **Response to Reviewer qGAg (1/3)**
>
> >**W1: The method appears to rely on both contrast and non-contrast images and this could limit the usability of the approach as not all use cases have both. Perhaps the authors could clarify the importance of this requirement and limitation in the paper?**
> - Thanks for the great comment. Although we design our model using contrast and non-contrast images, our model can be applied to an environment that has only contrast-enhanced images. To demonstrate this, we implement the proposed framework only using the angiography images and have added the results in Table 10 of Appendix C.6 of the revised paper.
> - As described in Resp_Table 2, the segmentation performance of our model without non-contrast background images is comparable or even superior to the baseline model. As the generation module of our DARL takes latent features of input images, our model can synthesize images involving the information of input angiography images based on semantic masks, and learn vessel features. This can be a unique characteristic compared to other semantic image synthesis models that typically take the images directly.
>
> ***Resp_Table 2.*** Ablation study on the use of non-contrast images (Dice for segmentation performance).
> | Method | XCAD | 130 XCA | 30 XCA | DRIVE | STARE |
> |-------------------------|:--------:|:---------:|:-------:|-------:|-------:|
> | Ours (baseline) |  0.636 | **0.595** | 0.572 | 0.525 |  0.508  |
> | Ours w/o non-contrast image | **0.644** | 0.518 | **0.597** | **0.550** | **0.538** |

---

> ### Author Response · Authors · 2022-11-22
> **We are looking forward to your feedback.**
>
> Dear Reviewer qGAg,
>
> We sincerely appreciate your time and efforts in reviewing our paper, and your constructive comments. We would like to kindly remind you that we tried our best to respond to your concerns with additional experiments, etc. Could you please go over our responses and let us know if there are any remaining issues?
>
> Best regards,
>
> Authors

---

### Official Review · Reviewer_dTPf · 2022-10-26

**Confidence:** 4
**Correctness:** 3
**Technical Novelty And Significance:** 3
**Empirical Novelty And Significance:** 3
**Recommendation:** 6

**Clarity, Quality, Novelty And Reproducibility:**

Clarity: The paper is written clearly.

Novelty: The key idea appears novel to me. While the idea of buiding one base model with switchable componets to deal with multiple tasks is not unseen, doing so in the context of DDPM is new.

Reproducibility: The paper is easy to understand but may be not so easy to reproduce without the codes.

Quality: The paper is of high quality with good motivation, decent novelty, and solid experiments.


**Strength And Weaknesses:**

Strength:
Good writing and easy to follow. Decent novelty. Solid experiments（but with the below weaknesses)

Weaknesses:

The paper claims a fully self-supervised manner for learning the model. In fact, for the segmentation path A that takes an image as input and outputs the segmentation mask, the adversarial loss is used on the mask output. Practically speaking, this is a bit uncessary as the real segmentation masks are used in the loss function. For each real segmentation mask, it comes with a real input image. Therefore, it is possible to conduct supervised training at least for this path.  I am afraid that the good performances in fact arises from this part.

There are two experiments that can can be done to verify the effectiveness of the proposed method.
1) Change the path A to a fully supervised one while keeping the path B unchanges and then  compare with fully supervised segmentation approach
2) Change the output masks of path A to synthetic masks.

**Summary Of The Paper:**

The paper presents a DDPM based method for vessel image synthesis and segmentation. The idea is to train one base DDPM for both tasks, using a switchable SPADE as a means for incorporating the dfferences between these two tasks. Further, the DDPM is trained in a self-supervised manner. Experiments on the benchmark datasets, including both vessel and non-vessel datasets, demonstrate that the proposed method is effective and achieve better performances that competing SOTA methods.

**Summary Of The Review:**

Overall, the paper is of high quality with good motivation, decent novelty, and solid experiments. The experiments should be augmented with new results to make the paper even stronger.

---

> ### Author Response · Authors · 2022-11-13
> **Response to Reviewer dTPf**
>
> >**W1: The paper claims a fully self-supervised manner for learning the model. In fact, for the segmentation path A that takes an image as input and outputs the segmentation mask, the adversarial loss is used on the mask output. Practically speaking, this is a bit unnecessary as the real segmentation masks are used in the loss function. For each real segmentation mask, it comes with a real input image. Therefore, it is possible to conduct supervised training at least for this path. I am afraid that the good performances in fact arise from this part. There are two experiments that can be done to verify the effectiveness of the proposed method.**
> - We appreciate your constructive comments. The reviewer is kindly reminded that the proposed method is trained in a fully unsupervised manner, so we did not utilize any real segmentation masks of angiography images for training, rather we used randomly generated synthetic fractal masks. Hence, in the segmentation path (A), we compute the adversarial loss for distinguishing the generated vessel segmentation maps $\hat{\mathbb{s}}^{v}$ from the synthetic fractal masks ${\\mathbb{s}}^{f}$. To clarify the use of data, we have provided a detailed explanation of the training dataset for each path in Table 6 of Appendix B.
>
> >**W1-1: Change the path A to a fully supervised one while keeping the path B unchanged and then compare with fully supervised segmentation approach**
> - Thanks for the constructive comment. Although there are no labeled pairs in the training dataset, per the reviewer's suggestion, we have conducted two supervised segmentation experiments utilizing the 12 pairs of labeled validation data, which include 1) training our model through only the (A) path, and 2) training our model by giving the labeled data to the (A) path while keeping the (B) path. We have added the details of these experiments in the paragraph of “Supervised” of Appendix C.6 and reported the results in Table 10 in the revised paper.
> - In Resp_Table 1 that reports the results briefly, we can observe that the model of 2) shows superior performance for all the datasets compared to the fully supervised model of 1), which demonstrates the effectiveness of the (B) path of our proposed framework. Moreover, it is noteworthy that the performance of our self-supervised model surpasses the supervised approach only using the (A) path by large margins, even though ours is trained without any supervised data.
>
> ***Resp_Table 1.*** Study on model contribution to supervised approaches (Dice for segmentation performance).
> |            Model              |  XCAD  | 130 XCA | 30 XCA | DRIVE | STARE |
> |-------------------------|:--------:|:---------:|:-------:|-------:|-------:|
> | 1) Supervised: (A) path          | 0.590 | 0.404 | 0.534 | 0.497 | 0.458 |
> | 2) Supervised: (A)+(B) path   | **0.705** | 0.454 | **0.649** | **0.544** | 0.464 |
> | Self-supervised: Ours          | 0.636 | **0.595** | 0.572 | 0.525 | **0.508** |
>
> >**W1-2: Change the output masks of path A to synthetic masks.**
> - Thanks for your comment. We would like to kindly assure the reviewer that we have already designed our model to generate vessel masks similar to synthetic fractal masks. In other words, the reviewer's suggested method is the exact way how our approach learns vessel segmentation in the (A) path in an adversarial manner. By feeding the synthetic fractal masks  $\mathbb{s}^{f}$ to the discriminator $D_s$, our model is trained to generate the realistic segmentation masks $\hat{\mathbb{s}}^{v}$ while fooling the discriminator. In fact, this idea of utilizing synthetic masks is a key contribution that makes our method more practical in a label-hungry scenario.

---

> ### Author Response · Authors · 2022-11-22
> **We are looking forward to your feedback.**
>
> Dear Reviewer dTPf,
>
> We sincerely appreciate your time and efforts in reviewing our paper, and your constructive comments. We would like to kindly remind you that we tried our best to respond to your concerns with additional experiments, etc. Could you please go over our responses and let us know if there are any remaining issues?
>
> Best regards,
>
> Authors

---

### Author Response · Authors · 2022-11-13
**General comment by Paper 3075 Authors**

We thank the detailed comments from the reviewers that helped us in improving our paper. To address the comments, we have made the following major changes, which can be found in the revised paper.

- We have clarified the contribution of our method throughout the paper. As commented by the reviewers, we present a novel diffusion adversarial learning model that 1) uses the diffusion models for estimating latent features to represent vessel structures, 2) combines the diffusion model and adversarial model to generate images in real-time, and is robust to noises and various modalities, and 3) efficiently unifies both the semantic image synthesis and vessel segmentation tasks into a single generation module through switchable SPADE layers. We demonstrated these through experimental results.
- Thanks to the constructive comments from reviewers, we have analyzed our model contribution to various learning frameworks of supervised and semi-supervised methods. From the extensive experiments, we verify that our model not only provides pseudo labels or inputs but also achieves high segmentation performance even when there are a few labeled data. We have described these experiments in Appendix C.6.
- We have provided a detailed explanation of the amounts of data we used for our experiments in Appendix B. We would like to emphasize that the proposed framework is trained in a fully unsupervised manner so we do not use any labeled data for training.
- While some reviewers point out that the proposed method is hard to implement due to the lack of a source code, we would like to kindly remind the reviewers that we have already submitted the source code as supplementary material to ensure anonymity. We will disclose the code repository in the final version and the trained network parameters as well.

Typos and other errors have been corrected. More details about the revision can be found in the responses to each reviewer.

---

### Author Response · Authors · 2022-12-05
**The end of the discussion phase approaching**

Dear Reviewers,

Could you please go over our responses since we can discuss them with you only for this week? We have faithfully responded to your comments and did our best to provide additional experimental results per your suggestions. We sincerely appreciate your time and efforts in reviewing our paper, and your constructive and insightful comments.

Thanks,

Authors

---

### Decision · Program_Chairs · 2023-01-20

**Decision:**

Accept: poster

**Justification For Why Not Higher Score:**

This is not supported by the reviewer scores.

**Justification For Why Not Lower Score:**

All reviewers recommend acceptance (and did so from the start).

**Metareview: Summary, Strengths And Weaknesses:**

Summary:
This paper mixes a diffusion model with adversarial learning to obtain a self-supervised model for vessel segmentation. This paper had 4 votes of (6 - accept) from the reviewers from the start.

Strengths:
- Well written paper
- Novel use of diffusion models for self supervision
- Strong experimental validation

Weaknesses:
- The dependency on both contrast and non-contrast images could greatly limit the potential usecases of the model
- The clarity could be improved; several reviewers have questions about performance
- While computational efficiency is enhanced as an advantage of the model, the difference to previous work -- as can be seen from the rebuttal -- is not as large as one might expect

**Note From Pc:**

if the above contains the word "oral" or "spotlight" please see: "oral" presentation means -> notable-top-5% and "spotlight" means -> notable-top-25%. As stated in our emails, we are disassociating presentation type from AC recommendations